# Avalanches and edge-of-chaos learning in neuromorphic nanowire networks

Joel Hochstetter [1✉], Ruomin Zhu[1], Alon Loeffler[1], Adrian Diaz-Alvarez[2], Tomonobu Nakayama [1,2,3] & Zdenka Kuncic [1,2,4✉]

The brain's efficient information processing is enabled by the interplay between its neuro-synaptic elements and complex network structure. This work reports on the neuromorphic dynamics of nanowire networks (NWNs), a unique brain-inspired system with synapse-like memristive junctions embedded within a recurrent neural network-like structure. Simulation and experiment elucidate how collective memristive switching gives rise to long-range transport pathways, drastically altering the network's global state via a discontinuous phase transition. The spatio-temporal properties of switching dynamics are found to be consistent with avalanches displaying power-law size and life-time distributions, with exponents obeying the crackling noise relationship, thus satisfying criteria for criticality, as observed in cortical neuronal cultures. Furthermore, NWNs adaptively respond to time varying stimuli, exhibiting diverse dynamics tunable from order to chaos. Dynamical states at the edge-of-chaos are found to optimise information processing for increasingly complex learning tasks. Overall, these results reveal a rich repertoire of emergent, collective neural-like dynamics in NWNs, thus demonstrating the potential for a neuromorphic advantage in information processing.

[1] School of Physics, University of Sydney, Sydney, NSW, Australia. [2] International Center for Materials Nanoarchitectonics (WPI-MANA), National Institute for Materials Science (NIMS), Tsukuba, Ibaraki, Japan. [3] Graduate School of Pure and Applied Sciences, University of Tsukuba, Tsukuba, Ibaraki, Japan. [4] The University of Sydney Nano Institute, Sydney, NSW, Australia. ✉email: joel.hochstetter@sydney.edu.au; zdenka.kuncic@sydney.edu.au

The age of big data has driven a renaissance in Artificial Intelligence (AI). Indeed, the ability of AI to find patterns in big data could arguably be described as super-human; the human brain is simply not designed for brute-force iterative optimisation at scale. Rather, the brain excels at processing information that is sparse, complex and changing dynamically in time. The increasing prevalence of streaming data requires a shift in neuro-inspired information processing paradigms, beyond static Artificial Neural Network (ANN) models used in AI. The brain's unique capacity for adaptive, real-time learning is enabled by the complex interplay between its neuro-synaptic non-linear elements and a recurrent network topology[1], with information processing manifested through emergent collective dynamics[2].

Physically implemented neuro-inspired information processing has been demonstrated by nano-electronic device components that integrate memory and computation, enabling a shift away from the traditional von Neumann architecture. Resistive switching memories[3] are an important class of such devices known as memristors[4]. Their electrical response depends not only on applied stimulus, but also on memory to past signals[5], which can mimic the short-term plasticity and long-term potentiation of neural synapses[6]. Utilising memristive switches as artificial synapses in crossbar architectures[7] has shown great promise in realising physical implementations of popular ANNs such as feed-forward[8] and convolutional neural networks[9].

Self-assembled networks of metallic nanowires with memristive cross-point junctions[10–13] are an approach to move from rigid crossbar array architectures towards a more neural-like structure in hardware. Nanowire networks (NWNs) exhibit a small-world topology[14], thought to be integral to the brain's own efficient processing ability[15]. Electrochemical metallic filament growth through electrically insulating, ionically conducting coatings (e.g. metal-oxides[16], $Ag_2S$[10] or PVP[11]), enables memristive switching at the metal-insulator-metal junctions[11,17]. The interplay between memristive switching and a recurrent network topology promotes emergence of collective, adaptive dynamics, such as formation of conducting transport pathways that can be dynamically tuned[18,19], giving networks a biologically plausible structural plasticity[20]. These neuromorphic properties equip NWNs with unique learning potential, with applications ranging from shortest-path optimisation[21] to associative memory[22]. Additionally, the neuromorphic dynamics of NWNs may be exploited for processing dynamic data in a reservoir computing framework[23], as shown in both experimental[24,25] and simulation[26–28] studies.

It has been widely postulated that optimal information processing in non-linear dynamical systems may be achieved close to

a phase transition, in a state known as criticality[29]. Distinct phase transitions are associated with criticality, notably 'avalanche criticality'[30] and 'edge-of-chaos criticality'[31]. In avalanche criticality, a system lies at the critical point of an activity–propagation transition where perturbations to the system may trigger cascades over a range of sizes and duration, characterised by scale-free power-law distributions. In sub-critical systems, activity can only propagate locally. Super-critical systems exhibit characteristically large avalanches that span the system. Scale-invariant avalanches, concomitant with avalanche criticality, have been observed in neuronal cultures[29,32,33] and neuromorphic systems comprised of percolating nanoparticles[34]. In edge-of-chaos criticality, dynamical states lie between order and disorder and the system retains infinite memory to perturbations. Edge-of-chaos dynamics have been observed in cortical networks[35] and appear to optimise computational performance in recurrent neural networks[36], echo state networks[37] and random boolean networks[38].

The observation of $1/f$ power spectra in neuromorphic NWNs has led to the suggestion that they may be poised at criticality[12,39]. While necessary, $1/f$ spectra are insufficient for criticality as $1/f$ noise can be produced by a diverse range of processes without spatio-temporal correlations, including uncoupled networks of isolated memristors[40]. Here, we present evidence for avalanche criticality in memristive NWNs and show that a critical-like state occurs near a first-order (discontinuous) phase transition. We also present the first evidence for edge-of-chaos criticality in neuromorphic NWNs and demonstrate that information processing is optimised at the edge-of-chaos for computationally complex tasks. Our results reveal new insights into neuro-inspired learning, suggesting that in addition to non-linear neuro-memristive junctions, the adaptive collective dynamics facilitated by neuromorphic network structure is essential for emergent brain-like functionality.

## Results

### A physically motivated model for neuromorphic structure and function in nanowire networks.
PVP-coated Ag nanowires self-assemble to form a highly disordered, complex network topology (experiment: Fig. 1a; simulation: Supplementary Fig. 1). As a neuromorphic device, NWNs are operated by applying an electrical bias between fixed electrode locations across the network[12,22]. To gain deeper insight into the neuromorphic dynamics, a physically motivated computational model of Ag-PVP NWNs was developed. The model is briefly described here (see Methods for further details).

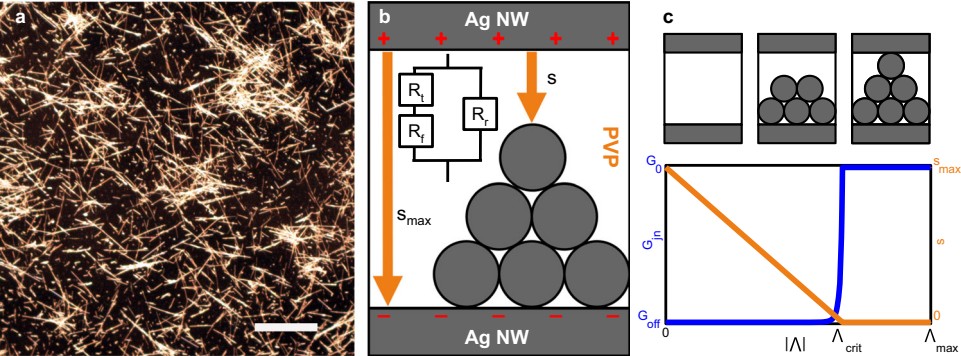

**Fig. 1 Ag-PVP nanowire networks with memristive junctions. a** Optical microscope image of a self-assembled Ag-PVP NWN (scalebar = 100 μm). **b** Biased nanowire-nanowire junctions promote $Ag^+$ migration and redox-induced Ag filament formation through the electrically insulating, ionically conducting PVP layer. Filament gap distance $s$ varies from $s_{max}$ to 0. Junction resistance is modelled as a constant series resistance ($R_r = R_{off} = G_{off}^{-1}$), in parallel with constant filamentary resistance ($R_f = G_0^{-1}$) and time-dependent tunnelling resistance ($R_t$). **c** Junction conductance ($G_{jn} = R_{jn}^{-1}$) and $s$, as a function of state variable $\Lambda(t)$. As $|\Lambda|$ increases, the junction transitions from high resistance to tunnelling and then ballistic transport.

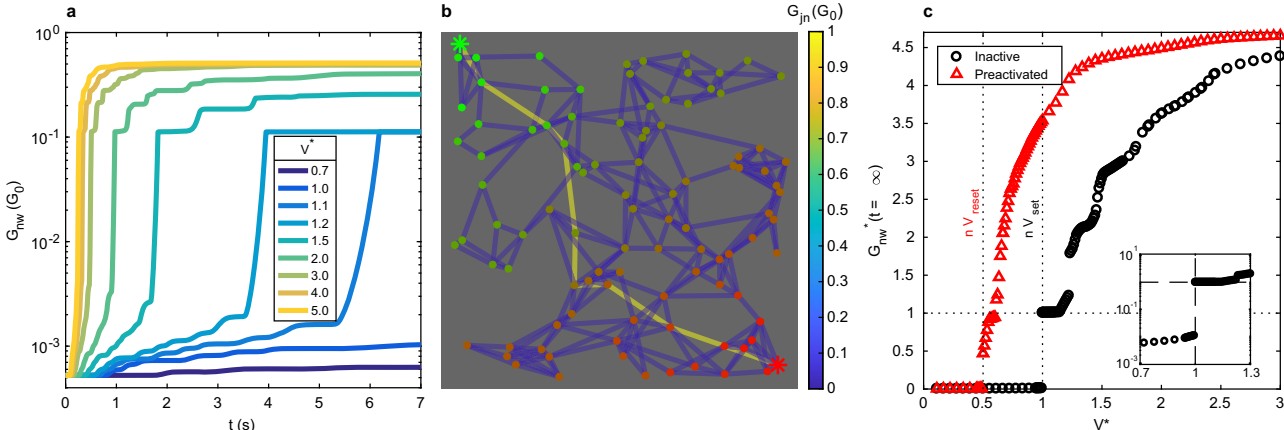

**Fig. 2 Threshold network activation under constant bias.** Simulations using a 100 nanowire, 261 junction NWN. **a** DC activation curves for an initially inactive NWN ($\Lambda = 0$ for all junctions) for increasing applied voltages ($V^* = V/V_{th}$), with $V_{th} = 0.09$ V. **b** Snapshot visualisation of NWN, showing formation of first transport pathway, corresponding to shortest path length $n$. **c** Steady-state network conductance (with $G^*_{nw} = nG_{nw}/G_0$), as a function of $V^*$. Black circles are for an initially inactive network. Red triangles are for a network pre-activated by a 1.8 V DC bias. Inset shows a zoom-in near $V^* = 1$ with a logarithmic vertical scale.

Ag|PVP|Ag junctions lie at the intersection between nanowires. These electrical junctions are modelled as voltage-controlled memristors[12,41]. $\Lambda(t)$ represents the state of a junction's conductive Ag filament in the PVP layer (Fig. 1b, Eq. (5)), and is closely related to the physical filament gap distance (Fig. 1b, c, Eq. (8)) that determines the junction's conductance ($G_{jn}(t) = R_{jn}^{-1} = I_{jn}/V_{jn}$).

A junction filament grows when voltage exceeds a threshold: $|V_{jn}| > V_{set}$. The sign of $\Lambda(t)$ encodes the junction polarity: a filament approaches $\Lambda = \Lambda_{max}$ when $V_{jn} > V_{set}$ and approaches $\Lambda = -\Lambda_{max}$ when $V_{jn} < -V_{set}$. When the voltage is sufficiently low ($|V_{jn}| < V_{reset}$), the filament decays towards $\Lambda = 0$, as a result of thermodynamic instability. The rate of filament growth or decay is determined by the voltage difference relative to $V_{set}$ or $V_{reset}$. Hence, $\Lambda(t)$ encodes a junction's memory to past electrical stimuli. See Methods (Eq. (5), Fig. 9) for full mathematical details.

Figure 1c shows the non-linear dependence of $G_{jn}$ on $|\Lambda|$ that produces switch-like junction dynamics. When $0 \leq |\Lambda| < \Lambda_{crit}$, the junction is insulating ($G_{jn} = G_{off} \ll G_{on}$, where $G_{on} = G_0$ is the conductance quantum). As $|\Lambda|$ approaches $\Lambda_{crit}$, the junction transitions to a tunnelling regime where conductance exponentially grows as $|\Lambda|$ increases. After the filament has grown ($\Lambda_{crit} \leq |\Lambda| < \Lambda_{max}$), the filament tip width is the limiting factor of conductance and transport is ballistic, with $G_{jn} = G_{on} = G_0$.

Next, simulations using this model are presented analysing the network level dynamics of this neuromorphic system.

**Adaptive network dynamics near a discontinuous phase transition.** NWNs driven by a constant bias $V$ converge to a steady-state conductance (simulation: Fig. 2; experiment: Supplementary Fig. 2a). The transient dynamics of the network conductance time-series, $G_{nw}(t)$, occurs in steps, as the network shifts between metastable states (plateaus in $G_{nw}$). Above a threshold voltage $V_{th}$, NWNs exhibit system-level switching, with $G_{nw}$ dramatically increasing by three orders of magnitude. Post-activation, the conductance increase slows to a rate comparable with pre-activation (on logarithmic scale). Increasing $V$ increases the activation rate and diminishes the step-like increases in conductance. These qualitative features of $G_{nw}(t)$ are independent of the specific network and voltage in simulation and experiment.

Simulations reveal that in the Low Conductance State (LCS: $G_{nw} \lesssim 10^{-3}$ $G_0$ for this network), all network junctions exhibit low tunnelling conductance and no junctions are in a ballistic

transport regime. The High Conductance State (HCS: $G_{nw} \gtrsim 10^{-1}$ $G_0$) exhibits pathways of high-conductance junctions ($G_{jn} \approx G_0$) spanning from source to drain electrodes. Figure 2b shows a single pathway, however the HCS may also consist of multiple, parallel high conductance pathways at higher voltages (Supplementary Fig. 3), and for large or dense networks with many equivalently short source-drain paths.

Figure 2c shows the steady-state $G_{nw}$ (re-scaled to $G^*_\infty = nG_{nw}(t = \infty)/G_0$, where $n$ is the graph length of the shortest path between source and drain electrodes) as a function of $V^* = V/V_{th}$ for two different network initial states: an inactive (LCS) network, with all junctions initialised to $\Lambda = 0$ (black circles); and a network pre-activated (HCS) to $G^*_{nw} = nG_{nw}/G_0 \approx 4.7$ by a 1.8 V $= 20V_{th}$ DC bias for 10 s (red triangles). For $V^* < nV_{reset}$, networks always converge to a stable LCS, irrespective of initial conditions. Above this level, multiple stable $G^*_\infty$ states emerge. Multi-stable conductance states for a wider range of initial conditions are shown in Supplementary Fig. 4. Networks exhibit hysteresis, since initial junction filament states control which stable state is reached. In simulation, for all initial conditions, $G^*_\infty$ is always non-decreasing with $V^*$, but this is not observed experimentally (Supplementary Fig. 2c) as it is difficult to prepare a network in exactly the same state before each activation (due to long-term memory of junctions). Additionally, persistent fluctuations from junction noise, not included in the model, can facilitate transitions between multi-stable states[12,39] (Supplementary Fig. 2b).

A discontinuity in the order parameter (steady-steady conductance, $G^*_\infty$) between LCS ($G^*_\infty \sim 0$) and HCS ($G^*_\infty \sim 1$) is revealed as the control parameter ($V$) is varied for any given network and initial conditions. This is found in both simulation (e.g. $V^* = 0.5$ for pre-activated and $V^* = 1$ for inactive networks in Fig. 2c) and experiment (Supplementary Fig. 2c). These distinctive discontinuities ($V^* = 0.5$ for pre-activated and $V^* = 1$ for inactive), along with the properties of multistability and hysteresis, indicate that the formation/annihilation of the first conducting pathway in the network corresponds to a first-order (discontinuous) non-equilibrium phase transition, marking an abrupt change in the global state of the system. In the model, the location of the discontinuous transition is universal (Supplementary Fig. 5), as it depends only on the length of the shortest path between the electrodes and on the initial state (either LCS or HCS). Networks in a LCS (no conducting pathways) transition to

a HCS (conducting pathways exist) when $V^* > nV_{set}$. Networks in a HCS transition to a LCS when $V^* < nV_{reset}$. In real (i.e. non-ideal) NWNs, $V_{set}$ and $V_{reset}$ may vary between junctions due to variations in the thickness of PVP coating around nanowires.

Additional discontinuities in $G^*_\infty$ may be observed at higher integer multiples of thresholds ($V = mV_{set}$, $V^* = kV_{reset}$). For example, at $V^* = 11/9 \approx 1.2$ (corresponding to $V = 11V_{set}$) in Fig. 2c. These correspond to the formation/annihilation of additional transport pathways. In the limit of large network size, only the first discontinuity (between $G^*_\infty \sim 0$ and $G^*_\infty \sim 1$) in $G^*_\infty$ vs $V$ remains concomitant with formation of the first conducting pathway (Supplementary Fig. 5).

These results demonstrate that NWNs can adaptively respond to external driving and can undergo a first-order phase transition between bi-stable states (LCS and HCS). These global network dynamical states arise from the recurrent connectivity between junctions and their switching dynamics, as shown next.

**Collective junction switching drives non-local transport.** Network activation or de-activation as described above can be understood as a collective effect emerging from recurrent connections between junctions. By Kirchoff's Voltage Law (KVL), the sum of voltages along any path between source and drain is equal to the externally applied voltage, while the sum of voltages around any closed loop in the network is zero. Since the voltage across a given junction $V_{jn}(t)$ controls the future evolution of its filament state $\Lambda(t)$ and hence conductance $G_{jn}(t)$, KVL couples the dynamics of memristive junctions to the network topology. A simulation at $V^* = 1.1$, with uniform initial conditions ($\Lambda = 0$), showing $G_{jn}(t)$ (Fig. 3a), $V_{jn}$ (t) (Fig. 3b) and corresponding visualisations at three time-points (Fig. 3c–e), reveals the qualitative characteristics of collective switching dynamics in NWNs.

Initially, the disordered network structure ensures the voltage distribution (Fig. 3b) is inhomogeneous, with junctions near the electrodes typically having the highest $V_{jn}$. For junctions with $|V_{jn}| < V_{set}$, filament states remain static, or otherwise decay to $|\Lambda| = 0$, if $\Lambda \neq 0$ and $|V_{jn}| < V_{reset}$. When $|V_{jn}| > V_{set}$, filaments grow at a constant rate $d\Lambda/dt = |V_{jn}| - V_{set}$ until $G_{jn}$ (Fig. 3a) rapidly increases from the onset of tunnelling, coinciding with a decrease in $V_{jn}$ (Fig. 3b). $G_{jn}$ plateaus once $|V_{jn}|$ reaches $V_{set}$. This first occurs at $t \approx 0.8$ s for junction #1. Voltage is redistributed locally to neighbouring junctions[42] (junctions near opposite electrodes, e.g. #1 and #9, can be considered connected by an edge, with $R_{jn} = 0$, at the power source). If the subsequent voltage increase results in $|V_{jn}| > V_{set}$, the filament growth rate increases for these junctions. For junctions already in a tunnelling regime ($G_{off} < G_{jn} < G_{on} = G_0$), $G_{jn}$ briefly increases until $V_{jn}$ returns to $V_{set}$, whereas insulating junctions experience a delay before $G_{jn}$ increases as their filament grows. Hence, $G_{nw}$ displays small, step-like increases coinciding with junctions transitioning from insulating ($G_{jn} = G_{off}$) to tunnelling ($G_{off} < G_{jn} < G_{on}$) and groups of already active junctions adaptively adjust their $G_{jn}$ such that their $V_{jn}$ lies on or below the threshold. These active junctions effectively wait for other junctions to activate before switching on, resulting in a switching synchronisation phenomenon similar to that seen in 1D memristive networks[43,44].

Through these switching events, the network self-organises to prevent a large voltage drop (greater than $V_{set}$) across any given junction, funnelling most of the current down a single (or a few) transport pathways, which grow from the electrodes towards the centre of the network (Fig. 3c, d). For any given pathway of length $m$, if $V^* < m V_{set}$, then after higher voltage junctions adjust their voltage to $V_{set}$ the pathway ceases to grow. This is the fate of all paths in networks with $V^* < 1$. When, $V^* > 1$, nearby pathways grow in competition, with the shortest pathway forming first.

Once the final junction along a source-drain pathway enters a tunnelling regime ($t \approx 6$ s), $G_{jn}$ of each junction on the path adjusts (Fig. 3e) such that they receive the same voltage ($V_{jn} = V/n > V_{set}$), since a larger $V_{jn}$ increases $d\Lambda/dt$ and hence, $G_{jn}$, which in turn reduces $V_{jn}$. At this point, junctions along the path behave collectively like a single memristive junction under constant bias: for each of these junctions, $\Lambda$ grows linearly in time, leading to exponential growth in $G_{jn}$ and hence, $G_{nw}$. This manifests itself as a large, rapid increase in $G_{nw}(t)$. This growth ceases when all junctions along the path reach the ballistic transport regime ($\Lambda = \Lambda_{crit}$, $G_{jn} = G_0$), where they stably remain at later times. For higher voltages or denser networks, additional transport pathways may form, resulting in later steps in $G_{nw}$ (cf. Fig. 2a). For sufficiently high voltages (when $V_{jn} > V_{set}$ for all junctions along the path), the plateaus become less pronounced and conductance increases continuously with time, however synchronous switching is still observed (Supplementary Fig. 6).

These results show that the emergence of transport pathways can be attributed to coupling between the complex network topology and memristive junction switching. Cascades of activity are induced as junctions transition into the conducting regime, adaptively redistributing voltage to their neighbours. This collective switching activity near a phase transition is reminiscent of avalanche dynamics, investigated next.

**Avalanche switching dynamics.** Avalanches with scale-free size and lifetime event statistics—a hallmark of critical dynamics—have been observed in neuronal populations[29,32,33] and other neuromorphic systems[34,45]. For NWNs, the conductance time-series of each junction (from the model described above) is converted to discrete switching events by introducing a threshold for the conductance change $\Delta G_{jn}$ between adjacent time-points. A similar procedure is applied to experimental measurements of $G_{nw}$. As in neuronal data[32,33], events are binned discretely in time into frames (width $\Delta t$). This allows an 'avalanche' to be defined as a sequence of frames containing events preceded and followed by an empty frame. Avalanche size ($S$) is defined as the total number of events in the avalanche. Avalanche life-time ($T$) is defined as the number of frames in the avalanche. See Methods for full details.

Systems near criticality exhibit avalanche size $S$ and lifetime $T$ probability distributions ($P(S)$, $P(T)$) and average avalanche size ($\langle S \rangle (T)$) that follow power-laws (Eqs. (1), (2), (3)), with exponents obeying the crackling noise relationship (Eq. (4))[46]. The agreement of the independent estimates (Eqs. (3), (4)) of $1/\sigma vz$ more rigorously tests avalanches for consistency with criticality than mere power-laws, as power-law size and life-time distributions can be obtained by thresholding stochastic processes[47], but do not obey Eq. (4).

$$P(S) \sim S^{-\tau} \tag{1}$$

$$P(T) \sim T^{-\alpha} \tag{2}$$

$$\langle S \rangle (T) \sim T^{1/\sigma vz} \tag{3}$$

$$\frac{1}{\sigma vz} = \frac{\alpha - 1}{\tau - 1} \tag{4}$$

Both simulated and experimental NWNs stimulated with voltages close to the switching threshold ($V^* \approx 1$) exhibit power-law $P(S)$ (Fig. 4a, d) and $P(T)$ (Fig. 4b, e) (KS test, $p > 0.5$). In the simulated NWNs, the power-laws exhibit a break attributed to the finite-size of the network limiting avalanche propagation. As network size or density is increased (Supplementary Fig. 7), the slope of the power-law region remains unchanged, while the power-law break increases for $P(S)$, $P(T)$ and $\langle S \rangle (T)$. The break in

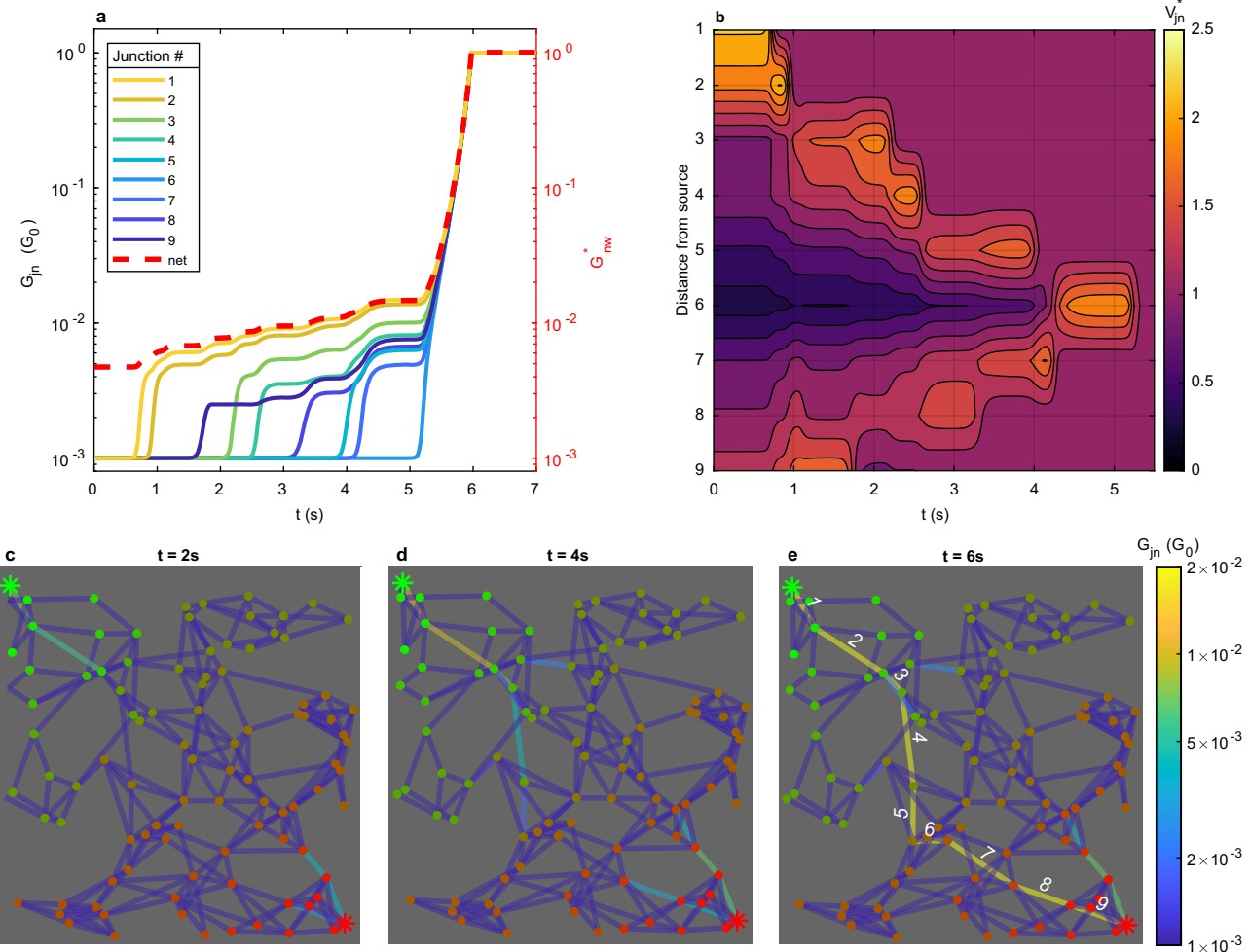

**Fig. 3 Collective switching dynamics under DC bias.** Simulations using a 100 nanowire, 261 junction NWN at voltage $V^* = V/V_{th} = 1.1$. **a** Conductance states of each junction, $G_{jn}$, along shortest source-drain path (in units of conductance quantum $G_0$). Junctions are numbered sequentially according to distance from source (#1 closest, #9 farthest). Network conductance, $G_{nw}^*(t)$ (normalised by source-drain path length), is also shown. **b** Corresponding junction voltage, $V_{jn}^* = V_{jn}/V_{set}$, dynamical re-distribution. **c–e** Snapshots of $G_{jn}$ distribution (edges shown on a logarithmic colourbar scale) at different time-points during network activation. Node colour indicates voltage on nanowires between source (green asterisk) and drain (red asterisk).

the distribution is consistent with criticality as the avalanche distributions obey finite-size scaling[48,49] (Supplementary Fig. 8a–c). Experimental NWNs contain many more nanowires so no power-law break is evident in the avalanches sampled. Additionally, $\langle S \rangle(T)$ follows a power-law (Fig. 4c, f). For simulated NWNs, $\tau \approx 1.95 \pm 0.05$ (Kolmogorov–Smirnov distance (KSD) = 0.006, $p = 0.54$) and $\alpha \approx 2.25 \pm 0.05$ (KSD = 0.004, $p = 0.54$), yielding $1/\sigma\tau z \approx 1.3 \pm 0.1$, which is in agreement with $1/\sigma\tau z \approx 1.3 \pm 0.05$ from $\langle S \rangle(T)$. The same agreement is obtained for experimental NWNs with $\tau \approx 2.05 \pm 0.10$ and $\alpha \approx 2.25 \pm 0.10$ (for $P(S)$, KSD = 0.032, $p = 0.80$; for $P(T)$, KSD = 0.037, $p = 0.59$), yielding $1/\sigma\tau z \approx 1.2 \pm 0.15$, which is in agreement with $1/\sigma\tau z = 1.2 \pm 0.05$ from $\langle S \rangle(T)$. The agreement of the individual estimates of exponent $1/\sigma vz$ in both simulated and experimental data confirms the crackling noise relationship (Eq. (4)) is obeyed within uncertainties. Further evidence for avalanche criticality is found by collapse of avalanche shape onto a universal scaling function (Supplementary Fig. 9), obtaining a third independent estimate that $1/\sigma vz \approx 1.3$. These factors strongly suggest that these avalanches are consistent with critical-like dynamics.

By altering the strength of the driving voltage away from the threshold $V_{th}$, the avalanche distributions begin to deviate from a power-law (Fig. 5). When $V^* < 1$, pathways are unable to form

across the network and the switching events result in small-scale avalanches (black points). As $V^*$ approaches 1, the distribution elongates and becomes a power-law (red points). Above $V^* = 1$ (when the networks activate), a bi-modal distribution is evident, as avalanches of large characteristic sizes and lifetimes emerge above the power-law tail (cyan and blue points). As network size increases, the probability density of the bump relative to the power-law region grows (Supplementary Fig. 8d–f). This suggests these anomalously large avalanches are consistent with super-critical states.

**Order-chaos transition from polarity-driven switching.** The response of a NWN to constant voltage stimulus, examined above, has revealed rich collective dynamics, but ultimately conductance converges to a steady state. When networks are driven by unipolar periodic stimuli, they converge to periodic attractor states. However, in response to a periodic, alternating polarity driving signal, NWNs exhibit a more diverse range of dynamics, from ordered to chaotic, depending on the amplitude $A$ and frequency $f$ of the driving signal. This is quantified by calculating the maximal Lyapunov exponent $\lambda$ for simulated networks driven by a triangular AC signal (see Methods).

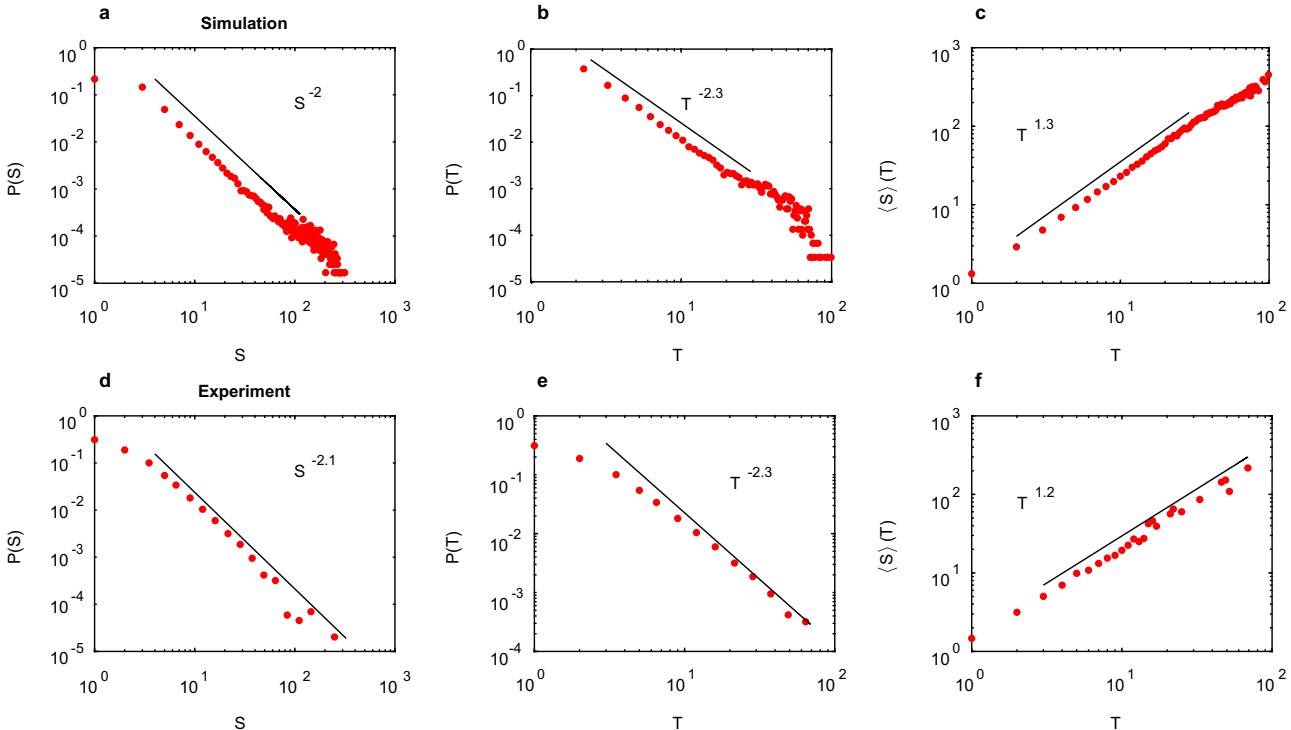

**Fig. 4 Avalanche statistics of simulated and experimental networks.** Top row—probability distributions of avalanche size $S$ (**a**) and life-time $T$ (**b**) and plot of average size as function of life-time (**c**) for simulated networks. Bottom row—probability distributions of avalanche size $S$ (**d**) and life-time $T$ (**e**) and plot of average size as function of life-time (**f**) for experimental networks. Statistics are produced from simulations on an ensemble of 1000 independently generated networks stimulated at $V^* = 1$. Simulations used networks with 2250 nanowires and 6800–7100 junctions (size $150 \times 150 \, \mu m^2$ and density 0.10 nw $(\mu m)^{-2}$). Experiments are at the same density and have size $500 \times 500 \, \mu m^2$. Maximum likelihood power-law fits are indicated.

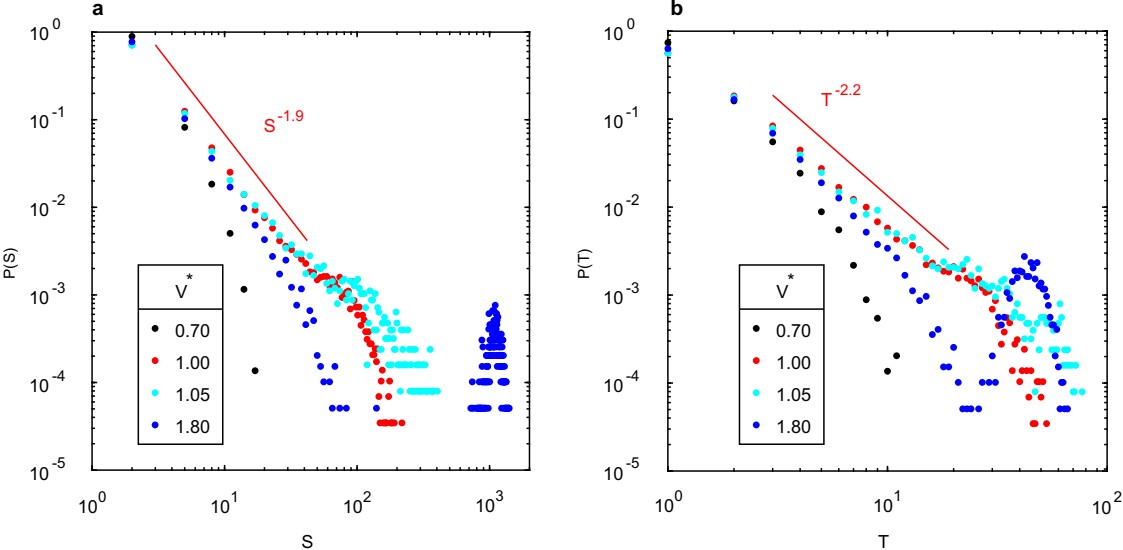

**Fig. 5 Voltage dependence of avalanche simulations.** Avalanche size $S$ (**a**) and life-time $T$ (**b**) distributions in simulated NWNs at voltages $V^* = 0.7$, 1.0, 1.05, 1.8 (black, red, cyan, blue). Statistics are produced from simulations on an ensemble of 1000 independently generated networks with 1000 nanowires and 3000–3200 junctions (size $100 \times 100 \, \mu m^2$ and density 0.10 nw $(\mu m)^{-2}$). The time-frame ($\Delta t = 160$ ms, corresponding to average inter-event-interval at $V^* = 1$) chosen to bin switching events is the same for each $V^*$ and distributions are linearly binned with same bin-sizes for each curve.

$\lambda$ measures the exponential convergence or divergence of nearby states of the system separated by an infinitesimal perturbation[50]. For $\lambda < 0$, the perturbations shrink over the trajectory and the network evolves to stationary (fixed point) or periodic (limit cycle) behaviour. For $\lambda > 0$, perturbations rapidly amplify,

resulting in chaotic dynamics. $\lambda \approx 0$ corresponds to the edge-of-chaos state.

Figure 6a plots $\lambda$ as a function of $A$ and $f$ for triangular AC stimuli. Highly ordered dynamics can be inferred from the low-$f$ and low-$A$ regime where $\lambda \lesssim -10 \, s^{-1}$ (black region). For $f \lesssim 0.2$ Hz,

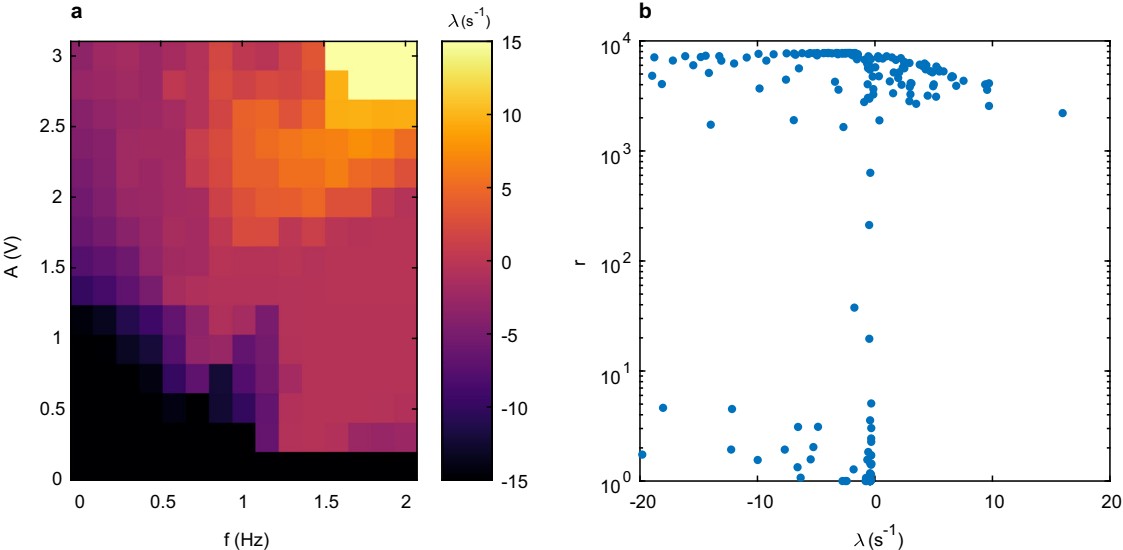

**Fig. 6 Controlling network dynamical states with triangular AC signals. a** For a 100 nanowire, 261 junction simulated network, the driving amplitude $A$ and frequency $f$ controls the maximal Lyapunov exponent $\lambda$ and hence, dynamical evolution of the network: $\lambda < 0$ indicates convergence to stable attractor states; $\lambda > 0$ indicates chaotic dynamics; and $\lambda \approx 0$ corresponds to an edge-of-chaos state. **b** For a given $\lambda$, the collective network response is captured by the average ratio of maximum-to-minimum network conductance states, $r$.

increasing $A$ results in an increase in $\lambda$ towards zero at $A \approx 3$ V. Above this frequency, increasing $A$ can lead to a $\lambda > 0$ and the onset of chaotic dynamics in the network. For $A \gtrsim 2.5$ V and $f \gtrsim 1.5$ Hz, stronger chaotic dynamics are inferred, with $\lambda \gtrsim 10$ s$^{-1}$ (yellow region). Further increasing $f$ for a given $A$ leads to a decrease of $\lambda$.

For a given $\lambda$, the collective switching states of NWNs was characterised by the ratio of maximum to minimum network conductance, $r$, averaged over each AC cycle. When $1 \lesssim r \lesssim 10$, the network remains in either LCS (no pathways) or HCS (pathways exist). When $r \gtrsim 10^3$, the network is persistently driven back and forth across the transition between these states. Figure 6b reveals that only networks exhibiting strong switching ($r \gtrsim 10^3$) can exhibit chaotic dynamics ($\lambda > 0$), whereas both edge-of-chaos dynamics ($\lambda \approx 0$) and ordered dynamics ($\lambda < 0$) are observed over the range of possible collective switching states.

Figure 7a–f shows simulated $I - V$ (top row) and $G - V$ (bottom row) curves over 20 successive triangular AC cycles for $A = 1.25$ V and varying $f$, producing different network dynamical regimes: ordered (left column, $f = 0.1$ Hz, $\lambda = -2.6$ s$^{-1}$); edge-of-chaos (centre, $f = 0.5$ Hz, $\lambda = 0.4$ s$^{-1}$); and chaotic (right column, $f = 0.85$ Hz, $\lambda = 4.1$ s$^{-1}$). Examples of corresponding junction conductance and voltage time-series are shown in Supplementary Fig. 10. In the $\lambda < 0$ regime, $I - V$ (Fig. 7a) and $G - V$ (Fig. 7d) cycles exhibit symmetric, repetitive hysteresis. For these states the minimum conductance occurs when $V = 0$, meaning the network fully activates and de-activates before polarity reverses. In the edge-of-chaos regime ($\lambda \approx 0$), $I - V$ (Fig. 7b) and $G - V$ (Fig. 7e) cycles are still symmetric and repetitive, but now the network does not deactivate completely before polarity reverses. In the chaotic regime ($\lambda > 0$), $I - V$ (Fig. 7c) and $G - V$ (Fig. 7f) trajectories diverge over successive cycles (evident from the thickening of blue curves). These chaotic trajectories are not unbound, but are instead confined to a reduced region of phase space (a 'chaotic attractor'). Examples of chaotic trajectories of different Lyapunov exponents are shown in Supplementary Fig. 11.

Qualitatively, the observed Lyapunov exponents can be understood by considering a small perturbation ($\delta\Lambda$) to the filament state ($\Lambda_i$) of the $i$-th junction. When tunnelling is absent

(cf. Fig. 1c), $\delta\Lambda$ changes $G_{jn}$ and hence, leaves the voltage distribution of the network unchanged. Thus, as the network continues to evolve, $\delta\Lambda$ is remembered by the junction (the junction states remain a fixed distance $\delta\Lambda$ apart). When $\Lambda_i$ approaches its boundaries (cf. Fig. 1), either from when the filament decays to zero ($\Lambda_i \rightarrow 0$ when $|V_i| < V_{reset}$) or saturates ($|\Lambda_i| \rightarrow \Lambda_{max}$ when $|V| > V_{set}$), $\delta\Lambda$ shrinks. When a junction is in a tunnelling regime, however, any perturbation to $\Lambda_i$ is exponentially amplified in terms of conductance. Perturbations that increase a junction's conductance decrease its voltage, slowing filament growth if $|V_i| > V_{set}$, or increasing filament decay if $|V_i| < V_{reset}$. The effect on neighbouring junctions is the opposite. Under slow driving, the network can adapt to such perturbations and retain the size of the perturbation. Under fast driving, however, the network is unable to adapt and perturbations grow during activation and de-activation, leading to separation of nearby network states. The frequency which constitutes fast or slow depends on the amplitude applied (cf. Fig. 6) and the network structure (size and density). The dynamical balance between the mechanisms that enforce order (perturbations shrink) and create chaos (perturbations grow) determines the stability of the global network dynamics. Hence, tuning the driving signal allows control over system dynamics. As shown next, this may be advantageous when utilising neuromorphic NWNs for information processing[36,45].

**Information processing optimised at the edge-of-chaos.** The information processing capacity of neuromorphic NWNs in different dynamical regimes was tested with the benchmark reservoir computing[51,52] task of non-linear wave-form transformation previously demonstrated in NWN experiments[24,25] and simulations[26,27]. In this task a triangular wave is input into the network, nanowire voltages are used as readouts and are trained using linear regression to different target wave-forms. Examples of target waveforms obtained are shown in Supplementary Fig. 12 and sample network readouts (nanowire voltages) are shown in Supplementary Fig. 13.

Figure 8 shows the network performance in transforming an initial triangular wave of given $A$ and $f$ to different target wave-forms as a function of the maximal Lyapunov exponent $\lambda$ for each

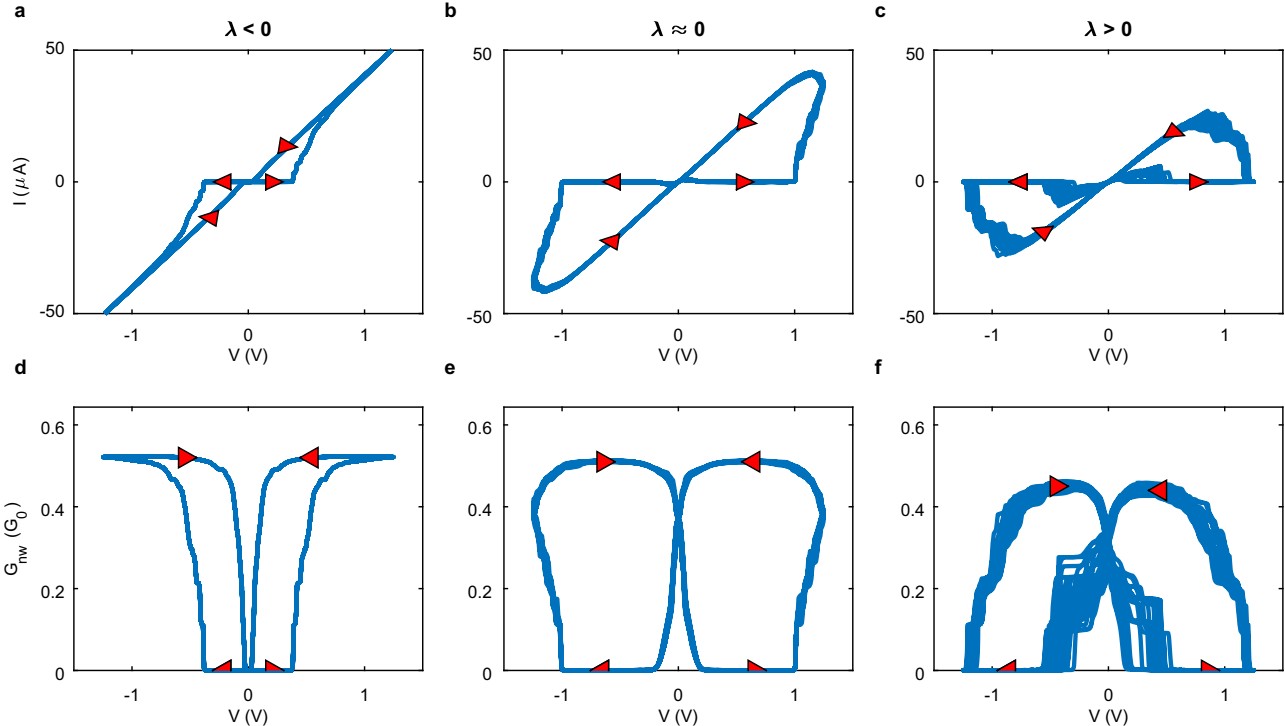

**Fig. 7 Ordered ($\lambda < 0$), edge-of-chaos ($\lambda \approx 0$) and chaotic ($\lambda > 0$) network states.** Each column corresponds to a single NWN simulation for a triangular AC input ($n = 20$) cycles, with $A = 1.25$ V and $f = 0.1, 0.5, 0.85$ Hz (left to right). Top row: $I - V$ curves, with arrows indicating direction of voltage control. Bottom row: corresponding network conductance, $G_{nw}$. First column (**a**, **d**): $f = 0.1$ Hz, $r = 7.7 \times 10^3$ and $\lambda = -2.6$ s$^{-1}$; all junction filament states return to 0 between cycles (reaching $V = 0$), resulting in symmetric repeatable $I - V$ and $G_{nw} - V$ cycles. Second column (**b**, **e**): states near the edge-of-chaos, with $f = 0.5$ Hz, $r = 7.1 \times 10^3$ and $\lambda = 0.4$ s$^{-1}$; network does not fully deactivate as polarity of voltage is reversed. Third column (**c**, **f**): $f = 0.85$ Hz, $r = 5.9 \times 10^3$ and $\lambda = 4.1$ s$^{-1}$; network trajectories are chaotic. A 100 nanowire, 261 junction network is used, but qualitatively similar results are found on a range of network sizes and densities.

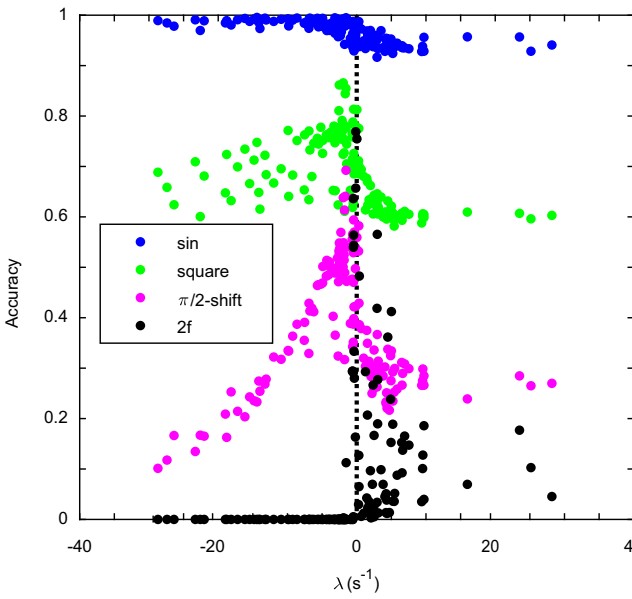

**Fig. 8 Network performance in non-linear transformation task for different dynamical regimes.** Accuracy of transformation of input triangular wave into different target waves: sinusoidal (blue), square (green), $\pi/2$ phase-shifted triangular (pink) and double frequency (2f) triangular (black), as a function of the network maximal Lyapunov exponent $\lambda$ (averaged over all junctions whose state changes while the task is being performed). Vertical line indicates $\lambda = 0$. Data-points correspond to individual simulations for a fixed $f$ and $A$ of the input wave for a 100 nanowire, 261 junction network.

input (cf. Fig. 6). Over the range of $\lambda$, the rank order of relative accuracy between tasks is consistent with the similarity between the input (triangular) and target waveforms, namely sin > square > $\pi/2$-shift ≈ 2f. The similarity between waveforms can be understood by considering the Fourier decomposition of each signal. To convert to a sine wave, higher harmonics must be removed from the triangular input signal, whereas conversion to a square wave requires additional odd higher harmonics. For double frequency conversion, odd higher harmonics must be removed, with even harmonics added. For phase shift conversion, the network must produce a lag to the input signal, i.e. coefficients of cosine terms in the Fourier series become coefficients of sine terms. The sine wave transformation accuracy is always $\geq 0.98$ before decreasing to $\approx 0.95$ above $\lambda \approx 0$. The square wave and $\pi/2$-shifted wave accuracy increase as $\lambda$ increases towards zero, peaking when $-2$ s$^{-1} \lesssim \lambda \lesssim 0$ s$^{-1}$ at 0.86 for square wave and 0.69 for $\pi/2$-shifted wave. Like the sine wave, both the square wave and $\pi/2$-shifted wave accuracy tapers off rapidly when the network is in a chaotic regime ($\lambda > 0$), to an approximately constant accuracy of ~0.6 for square and 0.3 for $\pi/2$-shift. Notably, for the $\pi/2$-shifted wave, the accuracy for chaotic states lies above the minimum accuracy for ordered states. For the 2f target wave, the accuracy is zero for $\lambda \lesssim -0.5$ s$^{-1}$, before rapidly increasing to peak near the edge-of chaos regime $\lambda \approx -0.1$ s$^{-1}$ at an accuracy of 0.79. Like the other wave-forms, accuracy decreases as the network becomes chaotic ($\lambda > 0$), but accuracy is higher than for ordered dynamics ($\lambda < 0$). For all tasks, strongly chaotic states underperform compared to the edge-of-chaos. This suggests that while more ordered (sine wave target) or slightly chaotic (2f target) dynamical regimes may be optimal depending on the computational complexity of the task, only a relatively

narrow range of network states, those tuned to the 'edge-of-chaos', are robust performers over a diverse set of target waveforms.

## Discussion

The formation (or destruction) of long-range transport pathways between electrodes is a ubiquitous feature of disordered memristive networks with threshold-driven junction switching[53], including nanowires[12,18,19] and nanoparticles[54]. Here, it was found that the network undergoes a discontinuous phase transition when the first pathway forms, with a linear relation between threshold voltage and source-drain path length. A similar discontinuous transition coinciding with pathway formation was reported in simulations of adiabatically driven memristive networks with large $G_{on}/G_{off}$ ratios (ratio between maximum and minimum possible $G_{jn}$)[53]. Smaller $G_{on}/G_{off}$ ratios were instead found to result in a continuous transition. When $G_{on}/G_{off}$ is large, changes to $G_{jn}$ lead to large changes in the voltage distribution, strongly coupling the dynamics of junctions in the network. This results in a switching synchronisation phenomenon (cf. Fig. 3a), analogous to that observed in 1D memristive networks[43,44]. Conversely, when $G_{on}/G_{off}$ is small, junctions changing conductance have a smaller effect on their neighbours. In this regime, junctions still collectively switch between metastable states, but do not exhibit switching synchronisation (Supplementary Fig. 14). Hence, by tuning $G_{on}/G_{off}$, which can be achieved experimentally by varying thickness of PVP coating, the collective behaviour of NWNs can be controlled.

Finding power-law avalanche distributions over a few orders of magnitude near a first-order (discontinuous) transition is surprising, as they are usually associated with second-order (continuous) transitions[55]. Disorder and hysteretic behaviour are likely key ingredients for scale-free avalanches observed in other systems driven near a discontinuous phase transition, such as magnetic materials[56] and neural networks[57]. These properties are certainly present in NWNs.

Critical avalanches were demonstrated here by tuning voltage to the activation threshold, but the role of self-organisation in achieving and sustaining critical avalanches requires further investigation. Experiments show NWNs continue to exhibit persistent, bi-directional conductance fluctuations (Supplementary Fig. 2b) as a result of junction noise, electromigration-induced junction breakdown events, and subsequent recurrent feedback by the network[12,58]. These mechanisms may allow NWNs to self-organise to an avalanche critical state, allowing mapping to models such as 'self-organised criticality' (SOC)[30] (self-organisation to a continuous transition), 'self-organised bistability' (SOB)[59] (self-organisation to a discontinuous transition), quasicriticality[60] (departs from criticality with crackling noise equation (Eq. (4)) obeyed) and their non-conservative counterparts[61]. Over long time scales, if anomalously large avalanches coexist with scale-free avalanches, then SOB would be a better description of NWNs than SOC. However, fluctuations could plausibly make the transition (cf. Fig. 2c) continuous resulting in a SOC-like state.

Critical dynamics has previously been observed in self-assembled tin nanoparticle networks (NPNs)[34,62]. There are notable differences between dynamics observed in nanowire and nanoparticle networks. In NWNs, resistive switching is facilitated by filament growth through an insulating layer. In NPNs, the nanoparticles are not coated with an insulating layer and resistive switching is due to tunelling-driven filament growth across nanogaps between nanoparticles. Thus, in the absence of filament growth, nanoparticles in contact are conductive, whereas nanowires in contact are insulating. Consequently, resistive switching

dynamics and critical avalanches are only observed when the nanoparticle density is finely tuned to the percolation threshold. NWNs, on the other hand, exhibit resistive switching and avalanches at densities close to and well above the percolation threshold, such as twice the threshold (cf. Supplementary Figs. 7 and 8). Conversely, in NPNs, the voltage does not need to be finely tuned to achieve critical avalanches, provided networks are on the percolation threshold. Breakage of conductive filaments from Joule heating and electromigration effects self-tune NPNs to a critical state. In NWNs, critical avalanches with power-law sizes and life-times are observed when tuning networks close to the threshold voltage. At voltages below the threshold, avalanches do not span the network and exhibit exponentially decaying avalanche distributions (cf. Fig. 5), while at voltages significantly above the threshold, large avalanches of a characteristic size and duration are observed, corresponding to formation and annihilation of non-local conducting pathways. Thus, passivation of nanoscale metallic components affords the advantage of not having to fine-tune density.

The universality of avalanches poses an interesting question for future studies on neuromorphic systems. The experimental studies of avalanches in percolating NPNs presented non-universal avalanche exponents that satisfy criteria for avalanche criticality[34,62]. Notably their model[62] exhibits avalanche exponents ($\tau = 2.0$, $\alpha = 2.3$, $1/\sigma \tau z = 1.3$) very close to NWN values, suggesting they may belong to the same universality class. A capacitive breakdown model of current-controlled NWNs, in a lower current regime than studied here,[63] found universal avalanche exponents consistent with the classic SOC sandpile model[30]. However, in other neuromorphic systems such as adiabatically driven memristive networks[53] and spiking neuromorphic networks[45], as well as neuronal culture experiments[32], avalanche statistics match that of a branching process[64], a member of the universality class of directed percolation ($\tau = 1.5$, $\alpha = 2$, $1/\sigma \tau z = 2$).

While ordered attractor ($\lambda < 0$) states in models of networks of voltage-controlled memristors under alternating polarity stimuli have previously been observed[65], the diverse edge-of-chaos and chaotic dynamics of neuromorphic NWNs have not been previously shown in memristive networks. Unlike certain types of memristors[66], individual junctions driven by periodic stimuli are incapable of exhibiting chaos, but chaos emerges in these networks due to strong recurrent coupling between components[67]. This requires an alternating polarity pulse where activation and de-activation are both strongly driven: for unipolar periodic pulses only $\lambda \leq 0$ is found. As the 'edge-of-chaos' ($\lambda \approx 0$) can be reached in a strongly driven regime (high $f$), it does not necessarily coexist with 'avalanche critical' states. Avalanche distributions ($P(S)$ and $P(T)$) near $\lambda \approx 0$ do not follow power-laws (Supplementary Fig. 15): power-law fits fail Kolmogorov–Smirnov test unless range of fit is made very small ($x_{max}/x_{min} \lesssim 2$), unlike the DC case at $V^* = 1$. This deviation may be attributed to the fast driving signal which ensures activity is injected into the network at a non-uniform rate while avalanches propagate, breaking time-scale separation between driving and network feedback (avalanches), obfuscating the distinction between consecutive avalanches. This reinforces the point often overlooked in neural network models that despite often coinciding[68], activity propagation (avalanche) and order-chaos transitions are distinct[69].

The observation of optimal overall performance on the non-linear transformation task at the 'edge-of-chaos' corroborates the popular hypothesis of robustness of information processing near phase transitions[31] and is consistent with simulations in other types of recurrent networks[36–38]. Despite this, the task dependence of accuracy is striking. For the simplest task

(transformation to sine wave) the 'edge-of-chaos' state afforded no computational advantage. On the other hand, the greatest pay-off from the 'edge-of-chaos' state was found for the most dissimilar target wave-forms (double frequency, phase shifted). This result corroborates a previous study using a spiking neuro-morphic network that found critical dynamics maximises the abstract properties of the system (auto-correlation time, susceptibility and information theoretic measures) and hence, performance in tasks of non-trivial computational complexity, yet for simpler tasks ordered dynamical states (away from criticality) may perform more optimally[45].

Neuromorphic NWNs may be utilised for a range of information processing tasks. Information may be stored in memristive junction pathways for static tasks such as associative memory[22] and image classification[26]. However, it is the coupling of memristive junctions with recurrent network topology that makes NWNs ideal for temporal information processing when implemented in a reservoir computing framework, such as signal transformation[24,25,27] and non-linear time-series forecasting[26,70]. These applications suggest NWNs are a promising neuromorphic system for adaptive signal processing of streaming data.

The rich dynamical behaviour revealed here may be observed on other network topologies, provided networks are highly recurrent and disordered. Recurrent networks have many short loops[71], allowing junctions to be strongly coupled (e.g. short loops in Fig. 2b which are coupled by Kirchoff's laws), thus generating the diverse range of time scales for avalanche events and chaotic dynamics to be observable. This recurrent topology is important for information processing in networks, such as recurrent neural networks[52,71]. Disorder ensures spatio-temporal inhomogeneity of the voltage distribution, maximising the degrees of freedom of networks. Understanding how to harness network structure for optimal information processing provides an exciting future challenge for neuromorphic network research.

In conclusion, neuromorphic NWNs respond adaptively and collectively to electrical stimuli and undergo a first-order phase transition in conductance at a threshold voltage determined by the shortest path length between electrodes. Due to recurrent loops in the network, junctions switch collectively as avalanches. These avalanches are consistent with avalanche criticality at the critical voltage. At higher voltages, anomalously large avalanches coexist with avalanches spanning a range of scales. Under alternating polarity stimuli, networks can be tuned between ordered and chaotic dynamical regimes. The edge-of-chaos is the most robust dynamical regime for information processing over a range of task complexities. These results suggest that neuromorphic NWNs can be tuned into regimes with diverse, brain-like collective dynamics[2,35], which may be leveraged to optimise information processing.

## Methods

**Simulations**. NWNs were modelled (Supplementary Fig. 1a) by randomly placing lines of varying length (randomly sampled from a gamma distribution with mean 10 μm and standard deviation 1 μm) and angular distribution (sampled uniformly on [0, π]) on an 30 × 30 μm² grid (centre locations sampled from a uniform distribution).

The network was converted to a graph representation (Supplementary Fig. 1b) in which nodes and edges correspond to equipotential nanowires and Ag|PVP|Ag junctions, respectively. The largest connected component was used in the simulations. Unless stated otherwise, a network with 100 nanowires and 261 junctions was used.

A voltage bias is applied between source and drain nodes (chosen at opposite ends of the network) and the model solves Kirchoff's laws to calculate the voltages $V_{jn}(t)$ across each junction at each time point. At each junction (edge), electrochemical metallisation is phenomenologically modelled with a conductive filament parameter ($\Lambda(t)$). The filament parameter is restricted to the interval $-\Lambda_{max} \leq \Lambda(t) \leq \Lambda_{max}$ and dynamically evolves according to a threshold-driven

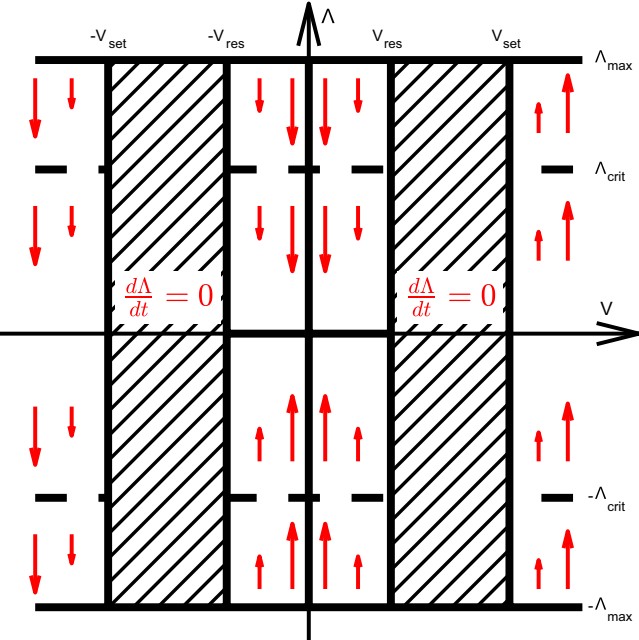

**Fig. 9 Pictorial representation of filament formation model.** Evolution of $\Lambda$ ($t$) (vertical axis) as a function of junction voltage $V_{jn}$ (horizontal axis) in threshold memristor junction model (Eq. (5)). Arrows are proportional to filament growth rate ($d\Lambda/dt$) and point in the direction of filament growth. When $V_{reset} \leq |V_{jn}| \leq V_{set}$, or $|\Lambda| = \Lambda_{max}$, $d\Lambda/dt = 0$.

bipolar memristive switch model (Fig. 9, Eq. (5))[41,72]:

$$\frac{d\Lambda}{dt} = \begin{cases} (|V_{jn}(t)| - V_{set}) \ \text{sgn}(V_{jn}(t)) & |V_{jn}(t)| > V_{set} \\ 0 & V_{reset} < |V(t)| < V_{set} \\ b \, (|V_{jn}(t)| - V_{reset}) \ \text{sgn}(\Lambda(t)) & V_{reset} > |V_{jn}(t)| \\ 0 & |\Lambda| \geq \Lambda_{max} \end{cases} \quad (5)$$

Junction conductance (cf. Fig. 1b, c) is modelled as a constant residual resistance of the insulating PVP layer ($R_r$), in parallel with constant filamentary resistance ($R_f = G_0^{-1} \ll R_r$) and $\Lambda$-dependent tunnelling resistance ($R_t$),

$$G(\Lambda) = \frac{1}{R_t(\Lambda) + R_f} + \frac{1}{R_{off}} \quad (6)$$

with tunnelling conductance $G_t$ calculated using the low voltage Simmon's formula (Eq. (7)) for a MIM junction[73].

$$\begin{aligned} G_t &= [R_t(\Lambda)]^{-1} \\ &= A \cdot \frac{3(2m_*)^{1/2} e^{5/2} (\phi/e)^{1/2}}{2h^2 s} \cdot \exp\left[-\frac{4\pi(2m_*)^{1/2}}{h} s \left(\frac{\phi}{e}\right)^{1/2}\right] \end{aligned} \quad (7)$$

$$s = \max\left[(\Lambda_{crit} - |\Lambda|) \frac{s_{max}}{\Lambda_{crit}}, \ 0\right] \quad (8)$$

with effective mass $m_* = 0.99 m_e$ and PVP layer (assumed homogeneous) thickness $s_{max} = 5$ nm. The potential barrier $\phi = 0.82$ eV is the difference between Fermi levels of PVP and Ag. $A = (0.41 \text{ nm})^2 = 0.17$ nm² is the area of a face of the silver unit cell. Nanowire resistance is considered negligible compared to junction resistance[11]. $G_t$ thus introduces an additional non-linear dependence on $V$, through the filament growth parameter $s = s(\Lambda(V))$, that modulates junction switching due to filament formation (cf. Supplementary Fig. 20).

Free parameters are chosen such that activation time is comparable to experiment. Values used are $V_{set} = 0.01$ V, $V_{reset} = V_{set}/2$, $\Lambda_{crit} = 0.01$ Vs, $\Lambda_{max} = 0.015$ Vs, $R_{off} = 10^3 (G_0)^{-1} = 12.9$ MΩ, $R_f = (G_0)^{-1} = 12.9$ kΩ and $b = 10$. For the Lyapunov analysis and non-linear transformation task $V_{set} = V_{reset}$ was used. Simulations use the Euler method with time-step $dt = 10^{-3}$ s. The effect of model parameters on results presented here is discussed in Supplementary Information.

**Experimental**. Physical NWNs were synthesised with the polyol process[74] using 1,2-propyleneglycol (PG) as an oxidising agent for silver nitrate (AgNO₃) and were drop-cast onto a glass substrate[12]. NWN size was 500 × 500 μm. Nanowires had mean length $7.0 \pm 2.4$ μm, diameter $500 \pm 100$ nm and density ≈ 0.1 nw/μm² determined with a high amplification optical microscope. The PVP-coating thickness is $1.2 \pm 0.5$ nm[12]. Networks were electrically stimulated with

pre-patterned rectangular gold electrodes of width 500 μm and current was read out using an in-house amplification system[12] at a sampling rate of 1 kHz.

**Avalanche analysis.** For simulation, an event is defined as when $\frac{1}{G_{jn}}|\frac{\Delta G_{jn}}{dt}|$ exceeds a certain threshold ($10^{-3}$ s$^{-1}$) before returning below the threshold. A similar procedure was applied to experimental data using the network level (rather than junction level) conductance time-series, with an event defined as when $\frac{1}{G_{nw}}|\frac{\Delta G_{nw}}{dt}|$ exceeds 1 s$^{-1}$ before returning below the threshold, or when $\Delta G_{nw}$, exceeds a threshold $5 \times 10^{-8}$ S before returning below the threshold. Changing the event detection threshold leads to negligible change of avalanche statistics.

As in studies of avalanches in neural cultures[32] and nanoparticle networks[34], the natural choice of frame ($\Delta t$) is the average inter-event-interval, $\langle IEI \rangle$, over the time-scale that events occur, where IEI is the time between adjacent events. Effect of changing frame on avalanche statistics is shown in Supplementary Fig. 16. Avalanche size and life-time distributions, $P(S)$ and $P(T)$, were binned linearly for simulations (logarithmically for experiments due to poorer resolution of the tail) and fit using a Maximum Likelihood (ML) power-law fit ($P(x) \propto x^{-\sigma}$). Scaling exponent ($\sigma$) were determined to precision $10^{-2}$, lower ($x_{min}$) and upper ($x_{max}$) cut-offs[75] were determined to nearest integer. Statistical significance of the ML fit was tested using the Kolmogorov–Smirnov (KS) test[76]. 500 synthetic data sets were generated from the distribution of the ML fit. The KS distance between each and the ML fit, was compared with the KS distance between the empirical (simulation or experiment) data and the ML fit. The $p$ value is the fraction of fits where the KS distance is smaller for the empirical data than the synthetic data. In all cases where power-laws were presented above, the null hypothesis that data followed a doubly truncated power-law was accepted to the chosen statistical significant ($p > 0.5$). For power-law fits with $p > 0.5$, no significant change was found to exponents (within uncertainties) for different $x_{min}$ and $x_{max}$ values up to a cut-off determined by system size (Supplementary Fig. 17). Hence, $x_{min}$ and $x_{max}$ are chosen such that $\log(x_{max}/x_{min})$ is maximised for fits with $p > 0.5$. As average avalanche size ($\langle S \rangle(T)$) is not in the form of a probability distribution (so ML methods do not apply), it was fit using linear regression on a log-log plot.

In simulation, as networks converge to a steady-state under constant DC bias (cf. Fig. 2), to generate avalanches, either a voltage pulse is applied to a network in a steady state (allowing certain junctions to activate/deactivate, subsequently triggering avalanches of other switching junctions), or junctions are manually perturbed by changing their state (e.g. switching them from high $G_{jn}$ to low $G_{jn}$ or vice versa). Results based on the former method are presented here. Avalanches are calculated from the transients as networks relax to their steady state using the binning method described above. To obtain enough statistics to sample the avalanche statistical distributions an ensemble of 1000 randomly generated networks of physical dimensions $150 \times 150$ μm$^2$ ($100 \times 100$ μm$^2$ for Fig. 5), density 0.10 nw (μm)$^{-2}$ and with nanowire length $10 \pm 1$ μm were simulated for 30 s each starting from all filament states from $\Lambda = 0$. All nanowires with physical centre location within 2.5% (3.75 μm for this network size) of top of network were selected as source electrodes. All nanowires with physical centre location within 2.5% of bottom of network were selected as drain electrodes. This ensured the whole network is stimulated, reducing finite-size effects.

In experiment, noise and junction breakdown events[12] perturb networks from their steady-state allowing spontaneous initiation of avalanches at constant voltage. NWNs were stimulated with DC biases of gradually increasing voltage (with waiting time of 3 h between measurements) until the switching threshold was determined. The network was then stimulated with a voltage just above threshold for 1000 s, or until current exceeded $8 \times 10^{-5}$ A (to prevent network damage), recording data at 30 kHz. This was repeated 3 times (with 3 h wait between measurements) on the same network. Time-series are truncated, first by excluding initial times for which network current $I < 1 \times 10^{-8}$ A and then between the first and last detected event. Avalanche statistics from all data-sets (at fixed voltage just above threshold) were combined to produce avalanche statistical distributions. Combining data-sets is valid under the assumption that avalanches at fixed voltage on an individual network follow the same statistical distribution.

Unlike in simulations, in experimental data, switching activity and avalanches in sub-threshold networks cannot be identified as they fall below the experimental noise floor. In contrast, junction conductance time-series can be simulated at all voltages. Examples of both experimental and simulated avalanches are shown in Supplementary Fig. 18.

**Lyapunov analysis.** A triangular wave of period $T$ was applied for a simulation time of 3000 s to allow the network to converge to an attractor. The perturbation method[50] was used to calculate the maximal Lyapunov exponent, $\lambda$. Briefly, the procedure was:

(1) Perturb filament state ($\Lambda_i$) of $i$-th junction in network by $\epsilon = 5 \times 10^{-4}$ Vs.
(2) Simultaneously evolve perturbed and unperturbed networks by one time-step ($dt = 5 \times 10^{-4}$ s) to obtain perturbed $\Lambda_p(t)$ (filament states for each junction) and unperturbed $\Lambda_u(t)$ state vectors.
(3) Compute Euclidean distance, $\gamma(t) = |\Lambda_p(t) - \Lambda_u(t)|$, and re-normalise perturbed state vector $\Lambda_p(t)$ to $\Lambda_u(t) + \frac{\epsilon}{\gamma(t)}(\Lambda_p(t) - \Lambda_u(t))$.
(4) Repeat steps 2 and 3 until end of period. Average over all time-steps to obtain junction Lyapunov exponent $\lambda_j(T) = \frac{1}{dt}\langle\ln(\gamma(t)/\epsilon)\rangle_t$.

(5) Repeat step 4 until $\lambda_j(T)$ converges to within error tolerance ($|\lambda_j(T) - \lambda_j(T - 1)| < 10^{-2}$ s$^{-1}$)
(6) Repeat steps 1–5 for all junctions. Average over $\lambda_j$ to obtain network Lyapunov exponent $\lambda = \langle\lambda_j\rangle$.

Note, Lyapunov analysis is only performed under periodic driving where network converges to time-varying attractor. Under constant DC voltage where network converges to a steady-state $\lambda$ is not as well defined.

**Non-linear transformation task.** Following the reservoir computing implementation analysed in previous studies[26,27], junction filament states were pre-initialised to the starting values used in Lyapunov analysis, and network was driven by AC triangular wave for 80 s. Target waves $T(t)$ were sine-wave, square wave, double frequency triangular wave, and triangular wave phase shifted by $\pi/2$. Absolute voltage ($v_i$ for $i$-th node) of each of the $n$ nanowires were used as reservoir readout signals, $\mathbf{V}(t) = [v_1(t), v_2(t), ..., v_n(t)]$, with corresponding weights $\mathbf{\Theta} = [\Theta_1, \Theta_2, ..., \Theta_n]^T$. The network's output signal is $y(t) = \mathbf{V}(t) \cdot \mathbf{\Theta}$. Weights were trained using linear regression to minimise the cost-function $J(\mathbf{\Theta}) = \sum_{m=1}^{M}(T(t_m) - y(t_m))^2$, over all time-points. Performance accuracy of the task is quantified by 1-RNMSE (Eq. (9)), where RNMSE is the root-normalised mean-squared error.

$$\text{RNMSE} = \sqrt{\frac{J(\mathbf{\Theta})}{\sum\limits_{m=1}^{M}[T(t_m)]^2}} \qquad (9)$$

## Data availability
Experimental data generated in this study have been provided as a Source Data file. Source data are provided with this paper.

## Code availability
Code to perform all simulations, process experimental data and generate all figures is on the repository https://github.com/joelhochstetter/NWNsim.

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

## Acknowledgements

The authors acknowledge use of the Artemis High Performance Computing resource at the Sydney Informatics Hub, a Core Research Facility of the University of Sydney.

## Author contributions

J.H. and Z.K. conceived and designed the study. J.H., Z.K., R.Z. and A.L. developed the model. A.D.A. synthesised the networks and performed the experiments. J.H. performed the simulations and analysed the simulation and experimental results. T.N. conceived the system. Z.K. supervised the project. J.H. wrote the manuscript, with consultation from other authors.

## Competing interests

The authors declare no competing interests.

**Additional information**

