## [Peer Review File · Nature Communications]

Reviewers' Comments:

Reviewer #1:

Remarks to the Author:

The manuscript points to interesting emergent and criticality phenomena found in electrically stimulated nanowire networks composed of memristive junctions. The work indeed contains remarkable insights about avalanche triggering and criticality in nanowire networks prone for neuromorphic response. However, I am not sure if this is the "the first evidence for avalanche criticality in memristive NWNs" as mentioned by the authors. Several citations presented by the authors (in the introduction) already pointed to such critical behaviour in nanowire networks.

One of my suggestions to the authors is to simplify the narrative of the manuscript as I find it very confusing. For example, it is not clear how the single-junction memristive model actually works. Is there a more transparent mathematical form for $d\Lambda/dt$? Figure 1d is not clear.

In addition, the main outcome of the manuscript (the avalanche distributions) needs extra clarification when it comes to the exact memristive model used to rule the network dynamics. The authors state "For NWNs, the continuous conductance time-series of each junction is converted to discrete switching events by considering the conductance change ΔG between adjacent time-points". So did the memristive dynamics now change to a binary frame in order to quantify discrete avalanches or are the authors using the same memristive model as illustrated in Figure 1c/1d but with a threshold that defines the switching? Which model is in fact exhibiting those scale-free dynamics?

I appreciate the work conducted by the authors and many of their conclusions will certainly benefit researchers working in this area. However, as I mentioned previously, I find the manuscript extremely confusing to read and to follow. The manuscript follows a too descriptive style and it is very dense. The main message of the work is diluted within too many details and specifications that make quite hard to follow. I would like to encourage authors to simplify the narrative of the manuscript in such a way to make it 'lighter' and more illustrative.

Reviewer #2:

Remarks to the Author:

See pdf attached below.

This manuscript describes simulations (and some experiments) for networks of silver nanowires which exhibit memristive behaviour due to the formation of filaments at the junctions between nanowires. These are interesting devices which are deservedly receiving a good deal of attention because they exhibit brain-like behaviour and have potential applications in neuromorphic computing. The scope of the work described is impressive, including investigations of multiple interesting effects related to the network dynamics, formation of pathways through the system, avalanching, order-chaos transitions and finally the optimisation of information processing at the edge of chaos. These are all interesting topics, but I found that their descriptions were too dense to allow a reader to really understand them, and that there were a considerable number of points that in any case needed clarification. These topics are clearly inter-related but because of the lack of clarity on the individual topics it was difficult to understand the inter-relationships or the overall validity of the conclusions. Furthermore, I found the relationship between the experiments and simulations to be unclear, and was also concerned that the considerable overlap with previously published work is also not made clear. While I have no doubt that there is significant novelty in this manuscript, what is novel and what has been previously published is difficult to discern.

I cannot recommend publication of this manuscript and would in fact encourage the authors to consider separating the content into 2 or 3 distinct articles which might do their interesting work more justice.

I have a large number of comments and questions and have only attempted to capture the key issues below.

1. The model of filament formation described in figure 1 and in the equations in the methods section is entirely reasonable but I could see no *evidence* that this model represents the process that actually takes place in the experiments. Is there in fact evidence of filament formation in their devices? How is this *specific* model justified? To what extent are the simulation results dependent on the model parameters? Previous results on silver nanowires (e.g. refs 10 and 36) described electrochemical processes and relaxation of the formed structures (and the manuscript describes relaxation in the experiments (over hours)) so why are these not included in the model? What are the impacts of these long relaxation times on the performance of real world devices? The timescale of the simulations is claimed to be ~ 7 s (Fig 2) - is this *real* time and if so how does it map to experimental timescales? It emerges later in the text that the authors consider there to be a transition for each junction from an insulating state to tunneling state but this is not discussed in the section about the model.
2. In several key cases the actual results from the simulations and experiments (Figs 4, 5 and 8) are not presented and insufficient detail of the analysis is presented to allow evaluation.
 - a. In the case of the avalanche analysis there is no discussion of the effect of changing the threshold for event detection, of the effect of cut-offs on the MLE analysis. If I understand correctly most of the data presented was for systems of 100 nanowires but even the data for 1000 nanowires in figure S7 it appears that the range of validity of the power law fits is very limited. Results of the KS test are not presented and there is no attempt to examine whether power laws provide the best fits. Most concerning is the absence of any data on shape collapse - this is fundamental to criticality analysis.

Given these concerns it is hard to accept the claim on page 7 that it is only data from a regular lattice that fails the KS test - a proper comparison of the NWN results with the regular arrays would strengthen the authors claims, as would a comparison with previously discussed network architectures (e.g. Refs 46, 53). The effect of system size (both number of wires and physical extent of the system) should be discussed. And is it really statistically valid to add data from all experimental data sets? (There is a version of the central limit theorem that says that adding independent datasets can spuriously result in power laws).

- b. In the case of the nonlinear transformation task, no data is presented showing the transformed signal. It is not possible to understand how the maximal λ averaged over all junctions relates to the distribution of λ s for the individual junctions, or why it is only the maximal value that matters (in fact this is a pretty typical approach in the literature, but the way this section is written does not make that clear). How the data varies with the various parameters is unclear – but the high degree of scatter in the data even after averaging and taking the maximum values raises obvious questions about reproducibility.

I did not understand the usefulness of the averaging in Fig. 8b – the average is completely dominated by the $\pi/2$ data so what does the averaging add? Is it possible to say why the network is better at producing some waveforms than others? The overlap with references 23 - 25 is not discussed: certainly the fact that nanowire networks can perform this task has been reported multiple times previously. The novelty appears to be that in this case results are presented as a function of λ but I wonder why the authors chose this task (which, as far as I am aware, has only been considered by one group previously and appears to be highly non-standard) when in references 24 and 25 more standard tasks such as the handwritten digit recognition task were presented. Finally, no experimental data is presented for this task which leaves the reader wondering again whether the simulations truly model the experiments. In the simulations it appears that outputs from every nanowire are trained - is it really plausible to do this in experiments?

3. I had detailed questions about other sections of the manuscript but list here just a few key concerns
 - a. The formation of apparently similar pathways between electrodes was discussed previously in Phys Rev E 92, 052134 (2015) as well as in references 16 and 17 (which is not cited anywhere in the section on pathways). There should be a discussion of the similarities and differences of the present work when compared with the literature. Is it really surprising that a pathway is formed between the 2 electrodes? And is there really a (singular) pathway - some of the figures show branches, so is this really a WTA process? If I understand correctly, the LCS and HCS are not in fact *states* because there are multiple levels with a range of conductances in each state - so what *exactly* do the terms LCS and HCS refer to, and does it really make sense to talk about a phase change? I found the discussion of discontinuities confusing: how do the authors distinguish between discontinuities due to a phase transition and discontinuities due to the switching of single junctions?

- b. the importance of the reversal of polarity was not clear to me (p8) which meant that the comments about the activation and collective effects (aren't these just reconfigurations/rewirings of the network?) were obscure. Am I correct in understanding that f alone governs λ ? If so, what exactly is the relationship between the two?
 - c. Is there *evidence* for recurrency and/ or re-excitation? While I understand that it is impossible to distinguish simultaneous avalanches in the experiments, this should be easy in the simulations.
4. A general concern is that there appears to be significant overlap with work on networks of nanoparticles, with several results appearing to be similar to those already published for those systems. The overlaps, and the advantages or otherwise of nanowire networks, should be discussed.

Why does criticality appear only close to the threshold voltage in NWNs but this does not appear to be the case in networks of nanoparticles? (And, as a side note, how is it even possible to measure avalanches (Fig. 5) below the voltage threshold?) In networks of nanoparticles the concept of percolation appears to be important, but percolation is not mentioned here (see also my comments above about the importance of the number of wires and size of the system, which should determine the percolation threshold).

5. There seems to be a certain amount of hyperbole at the expense of clarity e.g. is it really useful to emphasise Dragon avalanches but not say these are signatures of super-critical states? What is *optimal* about the paths – aren't they *just* paths? Similarly, why are the paths *topological*? Is information processing in the brain really *manifested* through emergent dynamics? The manuscript would be improved if there were less of these contentious points.
6. A few further details that should be attended to:
- a. the order of the presentation of the data e.g. why does the data in figure 7 presented after the results in figure 6
 - b. It would help the reader a lot if the first line of each caption made it clear whether the data is from simulations or experiments, and if the captions included all of the relevant details.
 - c. There are sections of the manuscript where there are hardly any references or explanations of key ideas e.g. the paragraphs about information processing at the edge of chaos.

Reviewer #3:

Remarks to the Author:

This is a very interesting paper that describes a range of experimental and modeling results on the resistance switching in random nanowire networks. Even though the system is quite complex and difficult to analyze, the authors have managed to extract a lot of information from dynamical data and explain their results using various concepts from non-linear dynamics and non-equilibrium thermodynamics. Of special interest is the study of optimal information processing capabilities of the system and its relation to the edge of chaos. It would be useful if the authors outline practical pathways for the utilization of the information processing capabilities of their system in real-world applications. Overall, the paper is well-written and scientifically sound. I have just a couple of minor comments.

Comments.

1. Section Results, second paragraph, "is mapped to a physical filament distance s (Fig. 1c)". Change "1c" to "1b"
2. Figure 1d. The model of filament growth needs to be clarified. Consider, for instance, Fig. 1d assuming that the filament is created at $V < V_{set}$ and then the voltage is abruptly changed to $V > V_{set}$. According to Fig. 1d, the filament is destroyed first (as λ goes to 0) and re-appears as λ goes to λ_{max} . Are there any data supporting such behavior?
3. Section Avalanche Switching dynamics. Please explain "dragon-king avalanches" in simple terms.
4. Information processing optimized at the edge-of-chaos. Please discuss any potential computing applications of the system.

We appreciate the detailed comments from the reviewers which have served to considerably improve the manuscript. Below we address all comments raised by the referees in a point-by-point manner.

Response to Reviewer #1:

The manuscript points to interesting emergent and criticality phenomena found in electrically stimulated nanowire networks composed of memristive junctions. The work indeed contains remarkable insights about avalanche triggering and criticality in nanowire networks prone for neuromorphic response.

However, I am not sure if this is the "the first evidence for avalanche criticality in memristive NWNs" as mentioned by the authors. Several citations presented by the authors (in the introduction) already pointed to such critical behaviour in nanowire networks.

We have revised the wording for this to "evidence for avalanche criticality in memristive NWNs".

One of my suggestions to the authors is to simplify the narrative of the manuscript as I find it very confusing. For example, it is not clear how the single-junction memristive model actually works. Is there a more transparent mathematical form for $d\Lambda/dt$? Figure 1d is not clear.

We have modified the qualitative description of the model in Results to make it more informative. A quantitative, mathematical description of the memristive model is presented in Methods, eqns. (2-5). This has been more clearly referenced in Results and Figure 1d has been moved to Methods (now Figure 9) to support the equations.

In addition, the main outcome of the manuscript (the avalanche distributions) needs extra clarification when it comes to the exact memristive model used to rule the network dynamics. The authors state "For NWNs, the continuous conductance time-series of each junction is converted to discrete switching events by considering the conductance change ΔG between adjacent time-points". So did the memristive dynamics now change to a binary frame in order to quantify discrete avalanches or are the authors using the same memristive model as illustrated in Figure 1c/1d but with a threshold that defines the switching? Which model is in fact exhibiting those scale-free dynamics?

We use the same memristive model as illustrated in Figure 1c/1d and eqns. (2-5) and apply a conductance threshold that defines switching events. The wording has been revised to explain this more clearly (p. 6):

"For NWNs, each junction's continuous conductance time-series (from the model described above) is converted to discrete switching events by applying a threshold for the conductance change ΔG_{jn} between adjacent time-points."

I appreciate the work conducted by the authors and many of their conclusions will certainly benefit researchers working in this area. However, as I mentioned previously, I find the manuscript extremely confusing to read and to follow. The manuscript follows a too descriptive style and it is very dense. The main message of the work is diluted within too many details and specifications that make quite hard to follow. I would like to encourage authors to simplify the narrative of the manuscript in such a way to make it 'lighter' and more illustrative.

We appreciate this valuable feedback. Significant revisions have been made to the manuscript to improve clarity of the writing and reduce density. Specific detailed sections have been broken up and re-written to remove details which dilute the main message of the work:

- Results, pp. 2-3: description of model has been re-written to more clearly explain the most important parts of the model; more technical details have been moved to methods.

- Results, p.7: discussion on relationship between network topology and avalanches has been greatly reduced. Avalanches on a square lattice result has been removed (we will perform a more detailed analysis of network topology in a future study).
- Results, pp. 7-9 the section “Order-chaos transition from polarity-driven switching” has been re-written to reduce dense, descriptive discussion on role of junctions in dynamics under periodic stimulation. Figures 7g-h have been moved to Supplementary information.
- Discussion, p. 11: detailed technical discussion around comparing avalanches in NWNs to branching process has been greatly reduced.
- Use of certain phrases that may lead to confusion such as “dragon-king avalanches” and “winner takes all” have been removed.

Response to Reviewer #2:

This manuscript describes simulations (and some experiments) for networks of silver nanowires which exhibit memristive behaviour due to the formation of filaments at the junctions between nanowires. These are interesting devices which are deservedly receiving a good deal of attention because they exhibit brain-like behaviour and have potential applications in neuromorphic computing. The scope of the work described is impressive, including investigations of multiple interesting effects related to the network dynamics, formation of pathways through the system, avalanching, order-chaos transitions and finally the optimisation of information processing at the edge of chaos. These are all interesting topics, but I found that their descriptions were too dense to allow a reader to really understand them, and that there were a considerable number of points that in any case needed clarification. These topics are clearly inter-related but because of the lack of clarity on the individual topics it was difficult to understand the inter-relationships or the overall validity of the conclusions. Furthermore, I found the relationship between the experiments and simulations to be unclear, and was also concerned that the considerable overlap with previously published work is also not made clear. While I have no doubt that there is significant novelty in this manuscript, what is novel and what has been previously published is difficult to discern.

I cannot recommend publication of this manuscript and would in fact encourage the authors to consider separating the content into 2 or 3 distinct articles which might do their interesting work more justice.

We appreciate this suggestion, but feel strongly that the work will have the greatest impact by presenting the neural-like dynamics (phase transition, avalanche criticality and edge-of-chaos learning) together. This is because these phenomena are interrelated and thus cannot be separated without significantly diminishing their overall significance.

We have, however, removed some stand-alone content (e.g. dynamics on a lattice), to be presented in future work.

I have a large number of comments and questions and have only attempted to capture the key issues below.

In addition to addressing these comments below, we have also made significant changes to the manuscript in response to comments from the other reviewers, specifically:

- We have removed some detailed descriptions (e.g., discussion of avalanches on lattice, comparison between avalanches in NWNs and branching process, role of junctions in AC dynamics) to be presented in future work.
- We have ceased use of confusing terminology (e.g. “dragon-king avalanches” and “winner-takes all”) that may lead to confusion for the reader.
- We have added a discussion on potential information processing applications of this neuromorphic system.

- 1. The model of filament formation described in figure 1 and in the equations in the methods section is entirely reasonable but I could see no *evidence* that this model represents the**

process that actually takes place in the experiments. Is there in fact evidence of filament formation in their devices? How is this *specific* model justified?

Direct visualization of filament formation across Ag-PVP-Ag nw-nw junctions is extremely challenging due to the size and density of the junctions. We do, however, have experimental measurements of single-junction I-V curves upon which we based our model, for example:

According to the literature, resistive switching (RS) across Ag-PVP-Ag junctions is most likely mediated by conductive filament formation (and dissolution) due to electrochemical metallisation:

- In Ag-PVP-Ag nanowire-nanowire junctions: Bellew et al. 2015 (ref 11), Manning et al. 2018 (ref 18), Milano et al. 2020 (ref 20).
- In PVP coated Ag nanoparticle films above the percolation threshold: Sandouk et. al 2015 (ref 17).
- Ag filament formation in Ag|PVP|Ag junction directly visualised by SEM in Yang et al. ACS Appl. Mater. Interf. (2020), <https://doi.org/10.1021/acsami.0c07533>

Similarly, in other Ag-insulator-Ag junctions:

- RS has been reported in junctions comprised of bare Ag-nanowires embedded in silk between gold electrodes, with molecular dynamics simulations showing filament formation: Wang et al. Nat. Commun. (2019), <https://doi.org/10.1038/s41467-018-07979-0>
- Ag filament formation in Ag|Ag-PEO|Ag junction directly visualised by SEM in Krishnan et al. Adv. Mater. (2016), <https://doi.org/10.1002/adma.201504202>
- Metallic nanowires with other types of insulating coatings have been found to exhibit RS by different mechanisms, e.g. migration of oxygen vacancies in the case of metal-oxide coating (<https://onlinelibrary.wiley.com/doi/abs/10.1002/aelm.201970044>) and a bias-catalysed phase change in the case of silver sulfide (<https://doi.org/10.1039/C5NR02536B>).

The evidence presented above for resistive switching attributed to conducting filament formation justifies our threshold memristive switching model of the Ag|PVP|Ag junction dynamics. As junctions are metal-insulator-metal systems with nanoscale dimensions, electron tunnelling cannot be ignored. Hence, the

inclusion of Simmons formula for tunnelling conductance. Our model explains the experimentally observed phenomenology of I-V curves: conductance increases above a voltage threshold in a highly non-linear manner and decreases both from instability of filaments (polarity independent) and reversal of bias (polarity dependent).

To what extent are the simulation results dependent on the model parameters?

We have performed parameter sensitivity analysis of these results that was not reported in the manuscript for brevity. The qualitative behaviour of the simulations does not have a strong dependence on model parameters. Each of the results presented are robust to changes to model parameters and similar phenomena can be seen, which can be attributed to the statistical properties of disordered networks.

V_{set} and V_{reset} affect the voltages at which individual junctions, and hence the network as a whole, activates or deactivates, respectively. When the ratio $V_{\text{set}}/V_{\text{reset}}$ remains fixed, identical behaviour is obtained by re-scaling the voltage applied to networks, and re-scaling time.

Λ_{crit} sets the time-scale for filament formation. Λ_{max} denotes the amount of memory of the junction to its high conductance state, increasing Λ_{max} increases memory of junctions and hence time-scale of their deactivation. Changing it leaves results unchanged. b scales the rate of filament decay compared to filament growth.

ϕ sets the tunneling barrier height. As ϕ increases, conductance approaches a step-function in Λ . As ϕ reduces, the exponential decay of filaments changes over a longer distance.

R_{off} and R_f set the scale of resistance of junctions and hence the networks. Provided $R_{\text{off}}/R_f \gg 1$ the dynamics of the networks is as shown in the paper. For lower R_{off}/R_f ratios, the “discontinuous transition” becomes continuous. This corroborates similar independent findings by Sheldon & Di Ventra (ref. 59) using a simplified model of a disordered memristive network. Some of the qualitative switching behaviour changes also as metastable states are not as clearly observed during DC activation. However, the focus of the paper is on the $R_{\text{off}}/R_f \gg 1$ regime as this corresponds to the physical networks.

For Lyapunov analysis: Changing b , the ratios $\Lambda_{\text{max}}/\Lambda_{\text{crit}}$ and $V_{\text{set}}/V_{\text{reset}}$ affect the frequencies and voltages at which chaotic, edge-of-chaos and ordered dynamics are reached. However, for each set of model parameters we explored the network can be tuned into these three regimes by sweeping frequency and amplitude of the driving signal. The same qualitative trend holds: above a certain amplitude increasing frequency causes Lyapunov exponent to first increase (from below to above $\lambda = 0$) then decrease (fall below $\lambda = 0$, for sufficiently high f).

Previous results on silver nanowires (e.g. refs 10 and 36) described electrochemical processes and relaxation of the formed structures (and the manuscript describes relaxation in the experiments (over hours)) so why are these not included in the model?

What are the impacts of these long relaxation times on the performance of real world devices?

ECM and relaxation are included in the model, parametrized by the filament parameter Λ , i.e. when $|V| < V_{\text{reset}}$, Λ decays towards zero resulting in a relaxation effect towards high resistance state of a given junction. The rate of relaxation is artificially sped up in the model to enable simulation and analysis of properties that would otherwise be difficult to perform experimentally. For example, our finding of a first-

order phase transition (cf. Fig. 2) from the simulation data requires days to replicate experimentally (c.f. Supp. Fig. 2c) because it requires the network to completely relax between measurements.

In real world devices, the long relaxation time means that after a network is activated, it takes a long period of time (hours-day) before the network resets to its initial state. This relatively long-term memory has been used to demonstrate performance in associative memory (ref. 22).

The timescale of the simulations is claimed to be ~ 7 s (Fig 2) - is this *real* time and if so how does it map to experimental timescales?

The timestep used in the model does not correspond to a *real* time interval, but is chosen such that the activation time roughly corresponds to that observed experimentally (Supp. Fig. 2a). Timescales differ even between experimental systems depending on the size of the network, length and density of the nanowires and thickness of the PVP coating of the nanowires. The nanowire networks simulated in this study are generally smaller than those studied experimentally. Thus, they can be considered scaled down versions of the experimental system.

It emerges later in the text that the authors consider there to be a transition for each junction from an insulating state to tunneling state but this is not discussed in the section about the model.

The transition from an insulating to tunnelling regime was mentioned in the first Results section describing the model (top of p.3 and also Fig. 1 caption). This section has been revised to be clearer and more explicit.

Previously:

“In this regime, tunnelling accounts for all current transport, with G_{jn} exponentially increasing as $s \rightarrow 0$.”

We added:

“When $0 \leq |\Lambda| < \Lambda_c$, the junction is insulating ($G_{jn} = G_{off} \ll G_0$, where G_0 is the conductance quantum). As $|\Lambda|$ approaches Λ_c , the junction transitions to a tunnelling regime where conductance exponentially grows as $|\Lambda|$ increases.”

Furthermore, this has been explicitly clarified later in the text. On page 5:

“coinciding with junctions transitioning from insulating ($G_{jn} = G_{off}$) to tunnelling ($G_{off} < G_{jn} < G_0$)”

2. **In several key cases the actual results from the simulations and experiments (Figs 4, 5 and 8) are not presented and insufficient detail of the analysis is presented to allow evaluation.**
 - a. **In the case of the avalanche analysis there is no discussion of the effect of changing the threshold for event detection, of the effect of cut-offs on the MLE analysis.**

We tested the effect on avalanche dynamics of changing the event detection threshold. This was omitted from the manuscript as there is no difference between distributions in the power-

law scaling regime of avalanches. There is negligible change to the distribution above the upper cut-off. Simulated avalanche distributions are shown below for different event detection thresholds (with 1×10^{-3} used in the results shown in figs. 4 and 5). The datapoints for 5×10^{-3} and below overlap entirely. We have added a comment to the Methods section:

“Changing the event detection threshold leads to negligible change of avalanche statistics.”

There was no significant effect of changing the cut-offs on the MLE analysis.

This is now briefly mentioned in the avalanche switching dynamics section of Results:

“For power-law fits with $p > 0.5$ there was no significant change to exponents (within uncertainties) for different x_{min} and x_{max} values.”

This can be clearly seen below in plots of τ , α and their corresponding p-values for different upper cut-off (x_{max}) with lower-cutoff fixed to $x_{\text{min}} = 5$. Before abrupt drop in p-value the corresponding exponents are the same within errors. We have changed the statistical significance of the power-law fitting to 0.5 in the manuscript to better account for abrupt change in goodness of fit.

Note: both the results above use an ensemble of 1000 simulated networks of size $100 \times 100 \mu\text{m}^2$ and density $0.10 \text{ nw}/\mu\text{m}^2$.

If I understand correctly most of the data presented was for systems of 100 nanowires but even the data for 1000 nanowires in figure S7 it appears that the range of validity of the power law fits is very limited.

As mentioned in Methods, the simulated avalanche data presented in the avalanche dynamics section of Results is for 1000 nanowires. The captions of figures 4 and 5 are updated to clearly state this. Further, we show avalanche distributions with 250 and 2250 nanowires in Supplementary Fig. 7.

Results of the KS test are not presented and there is no attempt to examine whether power laws provide the best fits.

The Kolmogorov–Smirnov distance and p-values of KS test have been added for each power-law presented in Results.

Following the prescription of Clauset 2009 (ref. 71) we compared fits for power-law (black), exponential (blue) and log-normal (pink) distributions. Visually the fits for exponential and log-normal distributions for each of these datasets is significantly worse than power-law fits. Statistical measures of goodness of fits corroborate this. For example, for experimental size distribution the log-likelihood found for power-law, exponential and log-normal are -5.1 , -4100 and -3900 , respectively. Log-likelihood ratios of ~ 1000 between power-law and other distributions are found for other datasets also. These simulated and experimental datasets correspond to those presented in Figure 4.

Most concerning is the absence of any data on shape collapse - this is fundamental to criticality analysis.

Data for shape collapse analysis has now been provided in Supplementary Fig. 8. Crackling noise exponent $1/\sigma_V$ is found to agree with results obtained by other methods within uncertainties. The collapse isn't perfect and can be improved by taking additional data. For both simulated and experimental data, the shape collapse is qualitatively comparable to that found in nanoparticle networks (refs. 33, 59).

Given these concerns it is hard to accept the claim on page 7 that it is only data from a regular lattice that fails the KS test - a proper comparison of the NWN results with the regular arrays would strengthen the authors claims, as would a comparison with previously discussed network architectures (e.g. Refs 46, 53).

We agree that a more comprehensive comparative analysis between avalanches on NWN network topology and regular lattices is required to draw firm conclusions. This has been removed from the manuscript.

The effect of system size (both number of wires and physical extent of the system) should be discussed.

This was shown in supplementary figure 7. This is now mentioned in Results, instead of Discussion, for further clarity. On page 6:

“The same power-law exponent, with an increasing power-law break is found by increasing the network size or density (Supp. Fig. 7).”

And is it really statistically valid to add data from all experimental data sets? (There is a version of the central limit theorem that says that adding independent datasets can spuriously result in power laws).

In both experiment and simulation, data from multiple time-series is combined. This is done to obtain enough statistics to produce the avalanche distributions, which would be unable to be resolved from a single experimental dataset. This is equivalent to sampling a different random instance of nanowire network structure in the ensemble of possible nanowire networks of a given size and density. If we were sampling networks which did not fall in the same class, then this would not be valid.

For example, the figure below shows experimental avalanche distributions of a single time-series from 2 different networks. The frame of each dataset is its average inter-event-interval. Each dataset passes KS test with $p > 0.5$, with power-law found to be best fit compared to exponential and lognormal (from log-likelihood ratio). Since the fitted exponent is most sensitive to the tail of the distribution, a more well resolved tail obtained from combining datasets within and between networks (Fig. 4) yields a more reliable exponent estimate than any single dataset.

Combining datasets of different networks (drawn from the same ensemble) has been used in experiments and simulations of avalanches in other systems to produce avalanche statistical distributions. For example, in simulations of capacitive breakdown of nanowire networks (ref. 60) and for some avalanche results presented in the seminal paper by Beggs and Plenz (ref. 31).

Together with the exponents (and shape collapse), the fact that the power law emerges only close to the phase transition (critical voltage) and does not persist far above or below is evidence for criticality and against it being a coincidental spurious feature.

b. In the case of the nonlinear transformation task, no data is presented showing the transformed signal.

Data showing an example of non-linear transformation to different target wave-forms has been added to the supplementary figures (Supp. Fig. 10).

It is not possible to understand how the maximal λ averaged over all junctions relates to the distribution of λ s for the individual junctions, or why it is only the maximal value that matters (in fact this is a pretty typical approach in the literature, but the way this section is written does not make that clear). How the data varies with the various parameters is unclear – but the high degree of scatter in the data even after averaging and taking the maximum values raises obvious questions about reproducibility.

The procedure does not involve “averaging and taking the maximum values”. A dynamical system has as many Lyapunov exponents as degrees of freedom, but here we are only interested in the maximal one as it determines whether the system is on a stable or chaotic attractor. The algorithm used, based off Sprott 2003 (ref. 47), directly estimates only the maximal Lyapunov exponent, as used widely in the literature including for Recurrent Neural Networks (e.g. <https://doi.org/10.1007/s12064-011-0146-8>, <https://doi.org/10.1016/j.neunet.2019.01.002>).

The results (order-chaos transition and accuracy of NLT task) are qualitatively reproducible on nanowire networks of different sizes. Furthermore, the variations between junction Lyapunov exponents do not significantly affect results presented in the manuscript. Fig. 8 is replotted below with error bars corresponding to standard deviation of junction Lyapunov exponents.

An example of the distribution of junction Lyapunov exponents is shown below. The exponents of junction Lyapunov exponents are found to take the same sign, in all but very particular cases. When the network Lyapunov exponent is close to zero, some junction Lyapunov exponents are positive, some are negative, but all close to zero.

I did not understand the usefulness of the averaging in Fig. 8b – the average is completely dominated by the $\pi/2$ data so what does the averaging add?

We agree that the average accuracy is not necessary and have removed it from the manuscript.

Is it possible to say why the network is better at producing some waveforms than others?

We previously stated (on page 9-10) that in general the network is better at producing wave-forms

more similar to the input wave-form.

“consistent with the similarity between the input (triangular) and target waveforms, namely $\sin > \text{square} > \pi/2\text{-shift} \approx 2f$ ”

We added the following to the Results section on page 9-10:

“The similarity between waveforms can be understood by considering the Fourier decomposition of each signal. To convert to a sine wave, higher harmonics must be removed from the triangular input signal, whereas conversion to a square wave requires additional odd higher harmonics. For double frequency conversion, odd higher harmonics must be removed, with even harmonics added. For phase shift conversion, the network must produce a lag to the input signal, i.e. coefficients of cosine terms in the Fourier series become coefficients of sine terms.”

The overlap with references 23 - 25 is not discussed: certainly the fact that nanowire networks can perform this task has been reported multiple times previously. The novelty appears to be that in this case results are presented as a function of λ but I wonder why the authors chose this task (which, as far as I am aware, has only been considered by one group previously and appears to be highly non-standard) when in references 24 and 25 more standard tasks such as the handwritten digit recognition task were presented.

We chose this task for several reasons: (i) the generation of higher harmonics used in this task is a defining property of recurrent neural networks; (ii) it has previously been demonstrated in an experimental nanowire network device (refs. 24, 25) and is thus relevant for real (non-ideal) networks; and (iii) it is relatively straightforward to implement and is thus ideal for studying relative performance in different dynamical regimes, which has not previously been attempted. By choosing this task, we have been able to directly test the hypothesis that learning performance is optimal at the edge of chaos, which was not considered in previous studies (refs. 24, 25, 26, 27). We could have chosen another task where we can select the level of complexity of the target signal, for instance forecasting the Mackey-Glass time series (refs. 26, 67), but the NLT task is certainly the simplest task that could be chosen to demonstrate relative performance in different dynamical regimes. Note the refs. 24, 25 are now refs. 26, 27.

Finally, no experimental data is presented for this task which leaves the reader wondering again whether the simulations truly model the experiments.

The non-linear wave transformation task was previously shown to work experimentally on a similar nanowire network using a multi-electrode array (ref. 24, 25). We did not attempt to perform experiments on this task, as we do not have the appropriate multi-electrode array to do so. Our electrodes are only at the edge of the network and provide few contact points, as opposed to contact points throughout the network. Note also that the objective of this study was not to determine performance per se, but to study relative performance for different dynamical regimes and it is less straightforward to control the dynamical regime of an experimental system compared to an ideal system (i.e. would require a lot of trial and error).

In the simulations it appears that outputs from every nanowire are trained - is it really plausible to do this in experiments?

We agree it is not practically feasible to train the output from every nanowire in an experimental system, where readout is limited by the number of electrode contacts.

However, as stressed above, the objective of this study is to show qualitatively how utilizing networks in different dynamical states can lead to vastly different task performance. Below we show that when the number of outputs is reduced from 100 (cf. Fig. 8) to 25 (5x5 electrode array), the trend in performance still holds, although now the edge-of-chaos advantage is only very clear for the computationally most complex cases (2f transformation and phase shift). We note that the experimental implementation of the NLT task reported in ref. 24 was performed with 16 electrodes and our experimental system has 18 electrodes (but arranged on opposite ends of the device), so the simulation result below gives some indication of what might be practically achievable.

The figure below shows accuracy as a function of number of rows in simulation with “electrode array” of outputs (ranging from 3x3 to 9x9 electrode array). Each curve corresponds to a particular amplitude and frequency pairing for the input signal. Increasing number of readouts, leads to improvement in task accuracy in each case.

3. I had detailed questions about other sections of the manuscript but list here just a few key concerns

- a. The formation of apparently similar pathways between electrodes was discussed previously in Phys Rev E 92, 052134 (2015) as well as in references 16 and 17 (which is not cited anywhere in the section on pathways). There should be a discussion of the similarities and differences of the present work when compared with the literature. Is it really surprising that a pathway is formed between the 2 electrodes?**

This reference has been added to the manuscript (ref. 51) and is now cited in the pathways section of Results. A more detailed discussion of the relation between new results and published literature has been added to the Discussion (p. 10):

“The formation (or destruction) of long-range transport pathways between electrodes is a ubiquitous feature of disordered memristive networks with threshold-driven junction switching (ref. 50), including nanowires (refs. 12, 18, 19) and nanoparticles (ref. 51). Here, it was found that the network undergoes a discontinuous phase transition when the first pathway forms, with a linear relation between threshold voltage and source-drain path length. A similar discontinuous transition coinciding with ‘pathway formation’ was reported in simulations of adiabatically-driven memristive networks with large R_{on}/R_{off} ratios (ref 50). Smaller R_{on}/R_{off} ratios were instead found to result in a continuous transition, which is also observed in simulations of NWNs (results not shown here). Experimentally, this could be tested using nanowires with thinner insulating coating (i.e. smaller junction R_{on}/R_{off}). The role of collective junction switching in pathway formation, analogous to switching synchronization in 1D memristive networks (refs. 42, 43), has not been identified before this study. This suggests that electron tunneling leads to richer dynamics than predicted by simpler binary switching models (refs. 12, 40).”

And is there really a (singular) pathway - some of the figures show branches, so is this really a WTA process?

The figures are plotted on a logarithmic conductance scale and are before network activation. The junctions along the branches are still at low conductance ($1/R_{off} < G_{jn} \ll G_0$). Once the first pathway forms, only junctions along this pathway have $G_{jn} \sim G_0$, while the remaining junctions are comparatively low conductance even though their filaments have partially grown ($1/R_{off} < G_{jn} \ll G_0$).

In the limit of large network sizes / dense networks, several pathways may form simultaneously, in which case it is not a winner-take-all process. As a result, we have removed WTA terminology from the paper and clarified this accordingly on page 3:

“Fig. 2b shows a single pathway, however the HCS may also consist of multiple, parallel high conductance pathways at higher voltages (Supplementary Fig. 3), and for large or dense networks with many short source-drain paths.”

If I understand correctly, the LCS and HCS are not in fact *states* because there are multiple levels with a range of conductances in each state - so what *exactly* do the terms LCS and HCS refer to, and does it really make sense to talk about a phase change? I found the discussion of discontinuities confusing: how do the authors distinguish between discontinuities due to a phase transition and discontinuities due to the switching of single junctions?

HCS refers to existence of non-local pathways spanning between electrodes and LCS refers to no such pathways. The LCS and HCS are global network states on either side of a phase transition. Although there are multiple levels (due to switching junctions), the LCS and HCS are stable to these fluctuations. This is analogous to, for example, a ferromagnetic-paramagnetic phase transition, on either side of which individual spins may continue to flip, but do not affect the global system state, which remains stable.

In our model, junction conductance is not discontinuous, but changes continuously with time (in contrast to binary models used in other studies). The large discontinuities in network steady-state conductance vs voltage are always found to correspond to the formation or annihilation of transport pathways spanning the network. In our model, as seen in Fig. 3, single junctions only switch on ($G_{jn} = G_0$) when they lie on a pathway which becomes conducting, otherwise their conductance plateaus at intermediate values ($1/R_{off} < G_{jn} < G_0$).

Considering formation of pathways in memristive networks as a phase transition has been discussed in previous studies (ref. 50). This process is similar to a percolation-like process occurring on networks when edges correspond to junctions in high conducting states ($G_{jn} \sim G_0$). In this sense the emergence of network spanning pathways can be considered a phase transition.

Additionally, there are global changes to network dynamics in the HCS. For example, emergence of large avalanches of characteristic size (super-critical avalanche distributions) corresponding to formation and destruction of pathways. These large avalanches do not occur in the LCS.

- b. the importance of the reversal of polarity was not clear to me (p8) which meant that the comments about the activation and collective effects (aren't these just reconfigurations/rewirings of the network?) were obscure.**

Under DC stimulation, we found only Lyapunov exponents less than or equal to zero, whereas under AC, exponents greater than zero were also found. Hence, polarity-driven switching enables access to dynamical regimes that are inaccessible under a DC bias.

In a sense there are reconfigurations/rewirings of the network, but these are not physical, rather a redistribution of conductance. Under AC, conducting pathways are continuously created and destroyed (whereas under DC, paths can only be destroyed if the bias is removed or reduced to below the threshold). Ordered dynamics corresponds to junctions consistently switching over adjacent cycles. Chaotic dynamics occurs when these cycles are too fast for the network to respond. In that case slightly different parts of the network become involved on different cycles.

The Results paragraphs on the role of polarity has been greatly simplified, with Fig 7(h-i) moved to the Supplementary Information (Supp. Figure 9).

- Am I correct in understanding that f alone governs λ ? If so, what exactly is the relationship between the two?**

No this is not correct as there is amplitude dependence on the Lyapunov exponent also, c.f. Fig. 6, which is presented in a different format in the figure below. Below a certain amplitude, only negative Lyapunov exponents are observed. Increasing the frequency causes the maximal Lyapunov exponent to first increase then decrease.

- c. Is there evidence for recurrency and/ or re-excitation? While I understand that it is impossible to distinguish simultaneous avalanches in the experiments, this should be easy in the simulations.**

Networks are recurrent due to many short loops ensured by the disordered nanowire network topology. We have investigated re-excitation in a single avalanche in simulation and could not find evidence for recurrency and re-excitation in the low voltage regime which exhibits power-law avalanches. However, as voltage is increased into the super-critical regime which exhibits anomalously large avalanches of characteristic size ($V^* = 1$), re-excitation of given junctions in an avalanche is observed. For clarity and brevity, we have removed the entire comparison between NWNs and branching process in the Discussion including the discussion of recurrency and re-excitation of avalanches.

4. **A general concern is that there appears to be significant overlap with work on networks of nanoparticles, with several results appearing to be similar to those already published for those systems. The overlaps, and the advantages or otherwise of nanowire networks, should be discussed.**

Why does criticality appear only close to the threshold voltage in NWNs but this does not appear to be the case in networks of nanoparticles?

(And, as a side note, how is it even possible to measure avalanches (Fig. 5) below the voltage threshold?)

In networks of nanoparticles the concept of percolation appears to be important, but percolation is not mentioned here (see also my comments above about the importance of the number of wires and size of the system, which should determine the percolation threshold).

We have acknowledged the results of nanoparticle networks in the Introduction (p.2) and Discussion (p. 11). A more detailed comparison between nanowire and nanoparticle networks has been added to the Discussion which addresses network density (concept of percolation) and differences in resistive switching (insulating coating vs no coating):

“Critical dynamics has previously been observed in self-assembled silver nanoparticle networks (NPNs) (refs. 33, 59). There are notable differences between dynamics observed in nanowire and nanoparticle networks. In NWNs, resistive switching is facilitated by filament growth through an insulating layer. In NPNs, the nanoparticles are not coated with an insulating layer and resistive switching is due to tunnelling-driven filament growth across nano-gaps between nanoparticles. Thus, in the absence of filament growth, nanoparticles in contact are conductive, whereas nanowires in contact are insulating. Consequently, resistive switching dynamics and critical avalanches are only observed when the nanoparticle density is finely tuned to the percolation threshold. NWNs, on the other hand, exhibit resistive switching and avalanches at densities close to and well above the percolation threshold, such as twice the threshold (cf. Supplementary Fig. 7). Conversely, in NPNs, the voltage does not need to be finely-tuned to achieve critical avalanches, provided networks are on the percolation threshold. Breakage of conductive filaments from Joule heating and electromigration effects self-tune NPNs to a critical state. In NWNs, critical avalanches with power-law sizes and life-times are observed when tuning networks close to the threshold voltage. At voltages below the threshold, avalanches do not span the network and exhibit exponentially decaying avalanche distributions (cf. Fig. 5), while at voltages significantly above the threshold, large avalanches of a characteristic size and duration are observed, corresponding to formation and annihilation of non-local conducting pathways. Thus, passivation of nanoscale metallic components affords the advantage of not having to fine-tune density.”

For the side note: Avalanches below the voltage threshold are only presented for simulation results, not experimental results.

5. **There seems to be a certain amount of hyperbole at the expense of clarity e.g. is it really useful to emphasise Dragon avalanches but not say these are signatures of super-critical states? What is *optimal* about the paths – aren't they *just* paths? Similarly, why are the paths *topological*? Is information processing in the brain really *manifested* through emergent dynamics? The manuscript would be improved if there were less of these contentious points.**

Each of the examples provided has a base in the literature, but we agree that changes are appropriate in some of these places to improve the clarity of the manuscript.

Dragon-king avalanches is a terminology originating from Sornette (<https://arxiv.org/abs/0907.4290>) and has been used in some recent neuronal avalanche literature (<https://link.springer.com/article/10.1140/epjst/e2012-01574-6>, <https://www.nature.com/articles/s41598-019-40473-1>). We agree the emphasis of Dragon-king avalanches may lead to confusion for the reader and detract from the main message of the manuscript. This has been modified to convey the sentiment suggested: “anomalously large avalanches are consistent with super-critical states”.

Optimal is terminology borrowed from Manning et. al 2018 (ref. 18) and Diaz-Alvarez 2019 (ref. 12) as conducting pathways are minimal energy connectivity pathways. However, this wording has been removed.

Topological path length in this sense refers to number of edges between two nodes, as opposed to a physical path length in nanometers. To avoid confusion that may arise between readers from different disciplines, this wording has been updated to “graph path length” in the first usage and simply “path length” after this.

6. A few further details that should be attended to:

- a. **the order of the presentation of the data e.g. why does the data in figure 7 presented after the results in figure 6**

Figure 6 introduces Lyapunov exponents and networks tunable from ordered to chaotic dynamical regimes. Whereas figure 7 provides examples of particular trajectories for different dynamical regimes. The order is chosen such that the concept of networks exhibiting different dynamical regimes is introduced before examples are shown of the particularities.

- b. **It would help the reader a lot if the first line of each caption made it clear whether the data is from simulations or experiments, and if the captions included all of the relevant details.**

Captions for figures 1-2, 4, 6, 7 already mentioned whether data corresponded to simulation or experiment. Captions for figures 3, 5 and 8 have been updated to indicate data corresponds to simulation. Further details have been added to the captions including network sizes for the simulated networks. In addition, we have updated all supplementary figures to specify whether results correspond to simulation or experiment and specify network sizes for simulations.

- c. **There are sections of the manuscript where there are hardly any references or explanations of key ideas e.g. the paragraphs about information processing at the edge of chaos.**

Additional references have now been added to key sections of the manuscript, including information processing at the edge of chaos.

Response to Reviewer #3:

This is a very interesting paper that describes a range of experimental and modeling results on the resistance switching in random nanowire networks. Even though the system is quite complex and difficult to analyze, the authors have managed to extract a lot of information from dynamical data and explain their results using various concepts from non-linear dynamics and non-equilibrium thermodynamics. Of special interest is the study of optimal information processing capabilities of the system and its relation to the edge of chaos. It would be useful if the authors outline practical pathways for the utilization of the information processing capabilities of their system in real-world applications. Overall, the paper is well-written and scientifically sound. I have just a couple of minor comments.

Comments.

1. Section Results, second paragraph, “is mapped to a physical filament distance s (Fig. 1c)”. Change “1c” to “1b”

This figure reference has been corrected.

2. Figure 1d. The model of filament growth needs to be clarified. Consider, for instance, Fig. 1d assuming that the filament is created at $V < -V_{set}$ and then the voltage is abruptly changed to $V > V_{set}$. According to Fig. 1d, the filament is destroyed first (as λ goes to 0) and re-appears as λ goes to λ_{max} . Are there any data supporting such behavior?

An example I-V curve based on experimental measurements of a single Ag-PVP-Ag junction is shown below under triangular voltage stimulus. The blue and red curves correspond to the first and second cycle respectively. The arrows indicate direction the cycles. A compliance current is used to ensure the junction does not exceed 1000 nA.

Starting at $V = 0$ mV, the junction activates at $V = 2000$ mV, until junction current reaches the compliance current. As voltage is reduced, the I-V curve is linear indicating fixed resistance high conductance state (referred to in our manuscript as the “ballistic transport regime”). Once the voltage is increased in the negative polarity direction, the junction deactivates (reduction in current). At $V = -2300$ mV the junction increases its current indicating a switch to high conductance state. At this point the voltage is ramped back towards zero and the cycle is repeated.

3. Section Avalanche Switching dynamics. Please explain “dragon-king avalanches” in simple terms.

Dragon-king avalanches refer to the bump above the power-law tail for avalanche size and lifetime distributions. This bump indicates large avalanches of characteristic size (and lifetime) occur in the network in these states. However, the terminology of “dragon-king avalanches” has been modified to remove confusion. We now refer to them as “anomalously large avalanches”. The focus on “dragon-king avalanches” has been reduced in the manuscript to simplify the narrative.

4. Information processing optimized at the edge-of-chaos. Please discuss any potential computing applications of the system.

Potential computing applications of the system are now explicitly stated in the Discussion (page 11-12), whereas previously only references were provided:

“Neuromorphic NWNs may be utilised for a range of information processing tasks. Information may be stored in memristive junction pathways for static tasks such as associative memory (ref. 22) and image classification (ref. 26). However, it is the coupling of memristive junctions with recurrent network topology that makes NWNs ideal for temporal information processing when implemented in a reservoir computing framework, such as signal transformation (refs. 24, 25, 27) and nonlinear time-series forecasting (refs. 26, 67). These applications suggest nanowire networks are a promising neuromorphic system for adaptive signal processing of streaming data.”

Reviewers' Comments:

Reviewer #1:

Remarks to the Author:

The authors made significant changes in the manuscript and addressed all points raised. I still find the narrative a bit dense but this may be due to the nature of the topic as well. In any case, the authors studied a rich current topic and the investigation is substantial and high quality. I am happy to recommend the work for publication.

NCOMMS-20-43196 second review
Reviewer #2

The authors have certainly improved the manuscript and I appreciate the detailed responses to my original questions - although I note that in some important cases corresponding changes were not made in the manuscript. The authors have simplified the structure and removed some contentious statements and I can now see that the manuscript might eventually be acceptable, but there are still a number of things which I found unclear or unsatisfactory. To keep the review short I will focus on those points.

Switching model. While the model of film growth is now much more clearly described in the new manuscript, the detailed model of switching appears to be motivated only by a reference to a rather obscure conference paper (ref. 40). This seems rather strange - I still think that some evidence that the model represents real experimental data (the author's own or from the literature) would strengthen the paper. The authors provide some evidence and references in their response to my original review and some of this material should certainly be included in their paper, along with some clarification of how the nonlinearities due to tunnelling can be separated from nonlinearities due to field growth. Similarly, in their response to my original question about dependence on model parameters, the authors provide a number of comments that should be included in the manuscript along with some supporting data. Given that there is essentially unlimited space in the supplementary information I do not agree that these points should be left out "for brevity".

MLE procedure/KS tests: I do not believe that the comment "For power-law fits with $p > 0.5$ there was no significant change to exponents (within uncertainties) for different x_{\min} and x_{\max} values." accurately reflects the data presented in response to my question about cut-offs. The reader needs to be aware that the range of x_{\max} for which acceptable values of p are obtained is actually quite small. It looks as though the upper limit corresponds to the position of the hump in the size and duration distributions i.e. the KS test is saying that the hump is not consistent with a powerlaw. I think it is important to show the effect of x_{\max} in the supplementary information and make a comment to the effect that this almost certainly shows that the system is slightly supercritical.

On a related point: some of the data presented (e.g. supplementary figure 7) suggests that the system is moving between subcritical, critical and supercritical states as a function of the number of nanowires. There should be a comment on this.

I accept that the adding of data from different datasets might be practically necessary, and that some other people have done the same thing in the past, but it is important that the manuscript acknowledges that this is a limitation of the analysis. The reality is that in adding the datasets together the authors are making the assumption that they all have the same distribution, but the only evidence for that claim is from the added datasets, which is a circular argument.

Growth of pathways: On p4: why does this change "mark an abrupt change in the *global state* of the system?" I understand the argument in the following paragraph which applies to *some* changes in conductance, but not all discontinuities are related to this collective effect. How is the reader to identify the collective effects? Many discontinuous changes are shown in figure 2 and it is still not clear to me which one is the authors considered to be "distinctive" and associated with the collective behavior. The author's response to my original question partially clarifies this, but additional changes to the manuscript are need here.

A related point: on P6: it is stated that conductance increases continuously in supplementary figure 6, but supplementary figure 6 looks very much like figure 3. The authors should clarify what they mean by discontinuous and continuous.

Topology: on p6 (and in several other places) the authors refer again to the importance of the complex network topology, but can they clarify what the topological effect is? Putting the question in another way: wouldn't the same arguments apply regardless of network topology? Is there anything special about the topology they are considering? I see that in the author's response to another reviewer they say that they removed discussion of topology from the manuscript – this means that claims that the topology is important are not supported by data.

Avalanches driven by different voltages. The discussion on p7 seems to be in conflict with the statements at the bottom of p5 where it is stated that the pathways cease to grow if $V^* < 1$. Presumably that means that avalanches completely die out? So how do they start up again to generate the data for $V^* = 0.7$ in figure 5? It would help enormously if the authors presented some examples of the raw conductance versus time data for the simulations. In essence: I think it is important to show what the avalanches actually look like.

A related point: in their response to my original question, the author's state that subcritical avalanches cannot be observed in the experiments but still do not explain how they can be observed in the simulations. A comment on this should be added to the manuscript.

Edge of chaos: Why are the different values of r (importance of which is highlighted in figure 6b) not evident in figure 7 or supplementary figure 9 (I understand that this shows *junction* resistances but surely these will lead to large r for all the networks shown here)? Perhaps this would be evident in logarithmic versions of figure 7d-e and these could be shown in additional panels in figure 7? Qualitatively the evidence for chaos in figure 7 seems limited: can the authors comment on why the trajectories shown in figure 7f vary from each other so little? The discussion of fast and slow driving on page 9 makes intuitive sense, but it would be useful to clarify what frequency ranges these correspond to.

Discussion: The new discussion and comparison with nanoparticle networks is interesting, as is the comparison between edge of chaos and avalanche criticality. My sense is however that that both this discussion and some of the details could be clarified.

- in the first paragraph, the importance of the on/off ratio is not really made clear. If the authors are saying that this somehow controlling the collective switching, then they should be more explicit.
- I could not see a clear statement of the value of λ in the avalanche critical state (e.g. figure 4). The author should surely explain how/why the DC and AC inputs lead to different avalanching behavior (Supplementary fig 12). In their response to my original question the authors state "Under DC stimulation, we found only Lyapunov exponents less than or equal to zero, whereas under AC, exponents greater than zero were also found." Something like this should be added to the paper, along with some commentary.
- on a related point: the $\lambda < 0$ data in supplementary figure 12 does not follow a straight line (i.e. appears to be subcritical). This suggests that the avalanche criticality exists between $\lambda = -2$ and 0, which in fact seems to be extremely close to the edge of chaos. Does this then invalidate the commentary in the discussion suggesting that the 2 types of criticality do not coexist? And in figure 8 the peaks of the green and pink curves appear to be just below $\lambda = 0$, suggesting maximum performance close to avalanche criticality, which further adds to the confusion.

A few smaller details:

P3: G_{nw}^* does not seem to be defined. Is it meant to be G_{nw} ?

P11: I checked references 33, 59 and the self assembled nanoparticle networks were made of tin not silver

Reviewer #3:

Remarks to the Author:

I am satisfied with how my questions have been addressed in the revised manuscript and, therefore, recommend its publication.

The authors have certainly improved the manuscript and I appreciate the detailed responses to my original questions - although I note that in some important cases corresponding changes were not made in the manuscript. The authors have simplified the structure and removed some contentious statements and I can now see that the manuscript might eventually be acceptable, but there are still a number of things which I found unclear or unsatisfactory. To keep the review short I will focus on those points.

Switching model. While the model of film growth is now much more clearly described in the new manuscript, the detailed model of switching appears to be motivated only by a reference to a rather obscure conference paper (ref. 40). This seems rather strange - I still think that some evidence that the model represents real experimental data (the author's own or from the literature) would strengthen the paper. The authors provide some evidence and references in their response to my original review and some of this material should certainly be included in their paper, along with some clarification of how the nonlinearities due to tunnelling can be separated from nonlinearities due to field growth.

Similarly, in their response to my original question about dependence on model parameters, the authors provide a number of comments that should be included in the manuscript along with some supporting data. Given that there is essentially unlimited space in the supplementary information I do not agree that these points should be left out "for brevity".

We have added new Supplementary Figures 14, 19-20 and a new Supplementary Notes section (p. 21, Supp. Info.) that includes model justification, parameter dependence and model assumptions.

Regarding tunnelling, the following sentence has been added to Methods, after eqn. (5): "G_t thus introduces an additional nonlinear dependence on V, through the filament growth parameter $s = s(\lambda(V))$, that modulates junction switching due to filament formation (cf. Supplementary Fig. 20)"

MLE procedure/KS tests: I do not believe that the comment "For power-law fits with $p > 0.5$ there was no significant change to exponents (within uncertainties) for different x_{\min} and x_{\max} values." accurately reflects the data presented in response to my question about cut-offs. The reader needs to be aware that the range of x_{\max} for which acceptable values of p are obtained is actually quite small. It looks as though the upper limit corresponds to the position of the hump in the size and duration distributions i.e. the KS test is saying that the hump is not consistent with a powerlaw. I think it is important to show the effect of x_{\max} in the supplementary information and make a comment to the effect that this almost certainly shows that the system is slightly supercritical.

On a related point: some of the data presented (e.g. supplementary figure 7) suggests that the system is moving between subcritical, critical and supercritical states as a function of the number of nanowires. There should be a comment on this.

We agree the range of the fit is limited, as stated on p. 7. The figure showing range of fits is added as Supplementary Figure 17. However, the existence of a hump in the avalanche size distribution does not necessarily imply super-criticality and may instead be attributed to finite-size effects, which is a more likely explanation given the small size of the networks simulated. See e.g. section 2.1.2.3 from Pruessner's book Self-Organized Criticality, aptly titled "The Hump" (<https://doi.org/10.1017/CBO9780511977671>).

Hence, both these points are resolved by binning avalanche size logarithmically to better resolve the distribution tail (in Supp. Fig. 7) and performing a finite-size scaling analysis (cf. Supp. Fig. 8a-c in revised Supp. Info.). For

each density at the critical voltage, the network obeys finite-size scaling. This indicates that at *each density* the network is critical, i.e. the apparent supercriticality is actually a finite size effect. The exception is at low density (just above the percolation threshold), when the network is sparser, with fewer recurrent feedback loops. Above the critical voltage, under attempts to apply finite size scaling, the height of the hump is found to diverge, indicating super-criticality (cf. Supp. Fig. 8d-f in revised Supp. Info.).

I accept that the adding of data from different datasets might be practically necessary, and that some other people have done the same thing in the past, but it is important that the manuscript acknowledges that this is a limitation of the analysis. The reality is that in adding the datasets together the authors are making the assumption that they all have the same distribution, but the only evidence for that claim is from the added datasets, which is a circular argument.

This assumption is now stated in Methods, p. 14, line 462: “Combining data-sets is valid under the assumption that avalanches from individual networks follow the same statistical distribution.”

Please note: the data provided in the previous response to reviewers is for single (not combined) datasets taken on individual networks. The avalanche distributions were compared to those of combined datasets presented in the manuscript.

Growth of pathways: On p4: why does this change “mark an abrupt change in the *global state* of the system?” I understand the argument in the following paragraph which applies to *some* changes in conductance, but not all discontinuities are related to this collective effect. How is the reader to identify the collective effects? Many discontinuous changes are shown in figure 2 and it is still not clear to me which one is the authors considered to be “distinctive” and associated with the collective behavior. The author’s response to my original question partially clarifies this, but additional changes to the manuscript are need here.

All discontinuities in the G-V plot are a collective effect as a result of groups of junctions switching. This has been clarified on p. 4, lines 126-130 of the revised manuscript.

A related point: on P6: it is stated that conductance increases continuously in supplementary figure 6, but supplementary figure 6 looks very much like figure 3. The authors should clarify what they mean by discontinuous and continuous.

This point has been clarified in the main text, p. 6, line 171: “increases continuously with time”. Discontinuous transition is used when only considering the steady state conductance vs. voltage.

Topology: on p6 (and in several other places) the authors refer again to the importance of the complex network topology, but can they clarify what the topological effect is? Putting the question in another way: wouldn’t the same arguments apply regardless of network topology? Is there anything special about the topology they are considering?

As it is important to clarify this point, a paragraph has been added to p. 12, line 376 of Discussion.

I see that in the author's response to another reviewer they say that they removed discussion of topology from the manuscript – this means that claims that the topology is important are not supported by data.

The particular result referred to is the study of avalanches on a square lattice. This was removed as the result was considered standalone to the results of the manuscript and hence dilutes the main ideas. This was in response to Reviewer 1's request to simplify the narrative and your (Reviewer 2's) suggestion to break up some information for future publications due to the broad scope of the manuscript.

Avalanches driven by different voltages. The discussion on p7 seems to be in conflict with the statements at the bottom of p5 where it is stated that the pathways cease to grow if $V^* < 1$. Presumably that means that avalanches completely die out? So how do they start up again to generate the data for $V^* = 0.7$ in figure 5? It would help enormously if the authors presented some examples of the raw conductance versus time data for the simulations. In essence: I think it is important to show what the avalanches actually look like.

A related point: in their response to my original question, the author's state that subcritical avalanches cannot be observed in the experiments but still do not explain how they can be observed in the simulations. A comment on this should be added to the manuscript.

Avalanches do indeed die out. For Fig. 5, data from different networks is combined. Both these points are now clarified in Methods (p. 14, line 442-447, 454-457) and in a new Supplementary Fig. 18 showing example avalanches.

Edge of chaos: Why are the different values of r (importance of which is highlighted in figure 6b) not evident in figure 7 or supplementary figure 9 (I understand that this shows *junction* resistances but surely these will lead to large r for all the networks shown here)? Perhaps this would be evident in logarithmic versions of figure 7d-e and these could be shown in additional panels in figure 7?

The values for r have been added to the captions of figure 7 and Supplementary Figures 10 and 11. Logarithmic versions of figure 7d-f curves are also now provided in Supp. Fig. 10.

Qualitatively the evidence for chaos in figure 7 seems limited: can the authors comment on why the trajectories shown in figure 7f vary from each other so little?

As stated on p. 8: "These chaotic trajectories are not unbound, but are instead confined to a reduced region of phase space (a 'chaotic attractor')." Supplementary Fig. 11 has been added showing other examples of chaotic trajectories that explore larger areas of G-V phase space.

The discussion of fast and slow driving on page 9 makes intuitive sense, but it would be useful to clarify what frequency ranges these correspond to.

Frequency ranges are shown in Fig. 6a and caption of Fig. 7. As there is some network dependence, a statement has been added to p. 9: "The frequency which constitutes 'fast' or 'slow' depends on the amplitude applied (c.f. Fig. 6) and the network structure (size and density)."

Discussion: The new discussion and comparison with nanoparticle networks is interesting, as is the comparison between edge of chaos and avalanche criticality. My sense is however that that both this discussion and some of the details could be clarified.

- **in the first paragraph, the importance of the on/off ratio is not really made clear. If the authors are saying that this somehow controlling the collective switching, then they should be more explicit.**

The connection with collective switching is now explicitly made in the Discussion on p. 10, lines 297-305, and Supplementary Figure 14 has been added.

- **I could not see a clear statement of the value of λ in the avalanche critical state (e.g. figure 4). The author should surely explain how/why the DC and AC inputs lead to different avalanching behavior (Supplementary fig 12). In their response to my original question the authors state “Under DC stimulation, we found only Lyapunov exponents less than or equal to zero, whereas under AC, exponents greater than zero were also found.” Something like this should be added to the paper, along with some commentary.**
- **on a related point: the $\lambda < 0$ data in supplementary figure 12 does not follow a straight line (i.e. appears to be subcritical). This suggests that the avalanche criticality exists between $\lambda = -2$ and 0 , which in fact seems to be extremely close to the edge of chaos. Does this then invalidate the commentary in the discussion suggesting that the 2 types of criticality do not coexist? And in figure 8 the peaks of the green and pink curves appear to be just below $\lambda = 0$, suggesting maximum performance close to avalanche criticality, which further adds to the confusion.**

Under constant DC, λ is not well defined as network converges to steady state (added to Methods, p. 14, line 477). Whereas under periodic DC stimuli, $\lambda \leq 0$ (added to p. 11, line 352).

For Supplementary Fig. 12 (now 15), we agree $\lambda < 0$ avalanche data appears sub-critical and $\lambda \geq 0$ super-critical, but there is no avalanche criticality in between. This is because the system is in the strong-driving regime. This has been clarified on p. 11, line 352-359 of Discussion and in the caption. Hence, it is not reasonable to infer from Fig. 8 that maximum performance is at avalanche criticality. A vertical line has been added in Fig. 8 to more clearly show performance is optimised at $\lambda = 0$ for the most computationally complex task (black data points), whereas $\lambda = 0$ becomes increasingly less of an advantage for less computationally complex tasks (pink, green blue data points).

A few smaller details:

P3: G_{nw} * does not seem to be defined. Is it meant to be G_{nw} ?

This is now defined at the first place it is introduced (p. 3, line 105)

P11: I checked references 33, 59 and the self assembled nanoparticle networks were made of tin not silver

This has been corrected.

NCOMMS-20-43196 third review
Reviewer #2

The authors have again improved the manuscript. I still feel the level of complexity and readability could be improved, but given that the other Reviewers have recommended acceptance, I am inclined to do likewise as long as a few particular issues that remain are addressed. In most cases a sentence or two would be sufficient.

- Switching model – while there is now a reasonable description of the model, it is still not clear how many of the results depend on the detail of the model. Can the authors add a comment on whether the dynamics of filament growth are compatible with the need to remain in the tunneling regime? Presumably tunneling occurs only over a very narrow range of gap distances and so is this realistic?
- Topology – despite the addition of a new paragraph in the conclusions (and the authors' latest responses), no evidence or references are provided to support the claims that topology is important and that recurrent loops exist.
- The new supplementary Fig 18 is helpful but raises further questions – how does the mean IEI for each example compare to the time scale shown? What do these avalanches look like after the binning procedure? The examples in a and b must surely have very low S and T and so it would be useful for the reader to understand how the values for the data in Fig 18 compare to the ranges shown in Fig. 4 and 5.
- The phrase “DC periodic” which appears several times in the manuscript. Perhaps “DC” is intended to mean “unipolar”?
- How can the conductance and current be zero at high voltage in fig S11? Normally G and I increase with V.
- I do not understand the authors' response to my question about the coexistence of avalanche and edge of chaos criticality. They respond that they do not co-exist “because the system is in the strong-driving regime” but the impact of strong driving regime on avalanche criticality is not obvious and requires explanation in the main text. This is an important point, and so the reasoning as to why coexistence can be ruled out needs to be clear (especially as the data seems to support the idea of coexistence).

The authors have again improved the manuscript. I still feel the level of complexity and readability could be improved, but given that the other Reviewers have recommended acceptance, I am inclined to do likewise as long as a few particular issues that remain are addressed. In most cases a sentence or two would be sufficient.

Switching model – while there is now a reasonable description of the model, it is still not clear how many of the results depend on the detail of the model. Can the authors add a comment on whether the dynamics of filament growth are compatible with the need to remain in the tunneling regime? Presumably tunneling occurs only over a very narrow range of gap distances and so is this realistic?

These points have been clarified in the Supplementary Notes on parameter dependence (p. 21): “The range of tunneling (determined by φ does not affect the observation of avalanches or chaotic dynamics. Further, it is not necessary to remain in the tunneling regime; all junctions which activate / deactivate will pass through this regime.”

Topology – despite the addition of a new paragraph in the conclusions (and the authors’ latest responses), no evidence or references are provided to support the claims that topology is important and that recurrent loops exist.

The paragraph on page 12 line 390-397 has been updated to highlight that recurrent loops exist: “(e.g. short loops in Fig. 2b which are coupled by Kirchoff's Law)”, and they are important for information processing: “This recurrent topology is important for information processing in networks, such as recurrent neural networks (refs 52, 71)”

The new supplementary Fig 18 is helpful but raises further questions – how does the mean IEI for each example compare to the time scale shown? What do these avalanches look like after the binning procedure? The examples in a and b must surely have very low S and T and so it would be useful for the reader to understand how the values for the data in Fig 18 compare to the ranges shown in Fig. 4 and 5.

The bin-sizes and mean inter-event-intervals are now included in the Supplementary Figure caption. Size and life-time distributions at different bin sizes were already included in Supplementary Figure 16.

The phrase “DC periodic” which appears several times in the manuscript. Perhaps “DC” is intended to mean “unipolar”?

“DC periodic” is now referred to as “unipolar periodic”.

How can the conductance and current be zero at high voltage in fig S11? Normally G and I increase with V.

The respective conductance and current are not zero ($G/G_0 \sim 10^{-3}$ - 10^{-1}). The network is inactive due to lag between junction filament growth and pathway formation across the network at high frequencies.

I do not understand the authors’ response to my question about the coexistence of avalanche and edge of chaos criticality. They respond that they do not co-exist “because the system is in the strong-driving regime” but the impact of strong driving regime on avalanche criticality is not obvious and requires explanation in the main text. This is an important point, and so the reasoning as to why

coexistence can be ruled out needs to be clear (especially as the data seems to support the idea of coexistence).

Co-existence is ruled out as avalanche distributions at high frequency (strong driving regime) cannot be fit to power-laws (fails KS test unless $x_{\max}/x_{\min} \lesssim 2$). This is now highlighted on p. 12 line 367. For comparison, $P(S)$ in the DC case (critical) and its corresponding best power-law fit has been added to Supplementary Fig 15.